**RESEARCH**

# Analysis of 1321 *Eubacterium rectale* genomes from metagenomes uncovers complex phylogeographic population structure and subspecies functional adaptations

Nicolai Karcher[1], Edoardo Pasolli[2], Francesco Asnicar[1], Kun D. Huang[1,3], Adrian Tett[1], Serena Manara[1], Federica Armanini[1], Debbie Bain[4], Sylvia H. Duncan[4], Petra Louis[4], Moreno Zolfo[1], Paolo Manghi[1], Mireia Valles-Colomer[1], Roberta Raffaetà[5], Omar Rota-Stabelli[3], Maria Carmen Collado[6], Georg Zeller[7], Daniel Falush[8], Frank Maixner[9], Alan W. Walker[4], Curtis Huttenhower[10,11] and Nicola Segata[1]*

* Correspondence: nicola.segata@unitn.it
[1]Department CIBIO, University of Trento, Trento, Italy
Full list of author information is available at the end of the article

## Abstract

**Background:** *Eubacterium rectale* is one of the most prevalent human gut bacteria, but its diversity and population genetics are not well understood because large-scale whole-genome investigations of this microbe have not been carried out.

**Results:** Here, we leverage metagenomic assembly followed by a reference-based binning strategy to screen over 6500 gut metagenomes spanning geography and lifestyle and reconstruct over 1300 *E. rectale* high-quality genomes from metagenomes. We extend previous results of biogeographic stratification, identifying a new subspecies predominantly found in African individuals and showing that closely related non-human primates do not harbor *E. rectale*. Comparison of pairwise genetic and geographic distances between subspecies suggests that isolation by distance and co-dispersal with human populations might have contributed to shaping the contemporary population structure of *E. rectale*. We confirm that a relatively recently diverged *E. rectale* subspecies specific to Europe consistently lacks motility operons and that it is immotile in vitro, probably due to ancestral genetic loss. The same subspecies exhibits expansion of its carbohydrate metabolism gene repertoire including the acquisition of a genomic island strongly enriched in glycosyltransferase genes involved in exopolysaccharide synthesis.

**Conclusions:** Our study provides new insights into the population structure and ecology of *E. rectale* and shows that shotgun metagenomes can enable population genomics studies of microbiota members at a resolution and scale previously attainable only by extensive isolate sequencing.

## Introduction

The composition of the human gut microbiota is variable across individuals and only few bacterial species are consistently present in populations of different geographic origin and lifestyle. Current large-scale metagenomic surveys [1] reported that merely three species (*Eubacterium rectale, Faecalibacterium prausnitzii, Ruminococcus torques*) and few other poorly characterized microbes are detected at > 0.1% relative abundance in more than 90% of adult healthy individuals [1]. In a recent study using metagenomic assembly and reference-independent binning, *E. rectale* was the species for which the most genomes from metagenomes could be reconstructed [2]. The large number of publicly available metagenomic cohorts and accurate methods for genome reconstruction from metagenomes thus provide an unprecedented opportunity to gain insights into this otherwise relatively poorly investigated bacterial species using metagenomic data at a global scale.

*E. rectale* is a member of the *Firmicutes* phylum, belonging to the *Lachnospiraceae* family. The proposed type strain of *E. rectale* (A1–86) is rod-shaped, Gram-positive, strictly anaerobic, and motile [3]. *E. rectale* produces butyrate and other short-chain fatty acids (SCFAs) from carbohydrates not directly accessible by the host, which play a role in promoting intestinal health in the host [4]. The relative abundance of *E. rectale* in the gut has been reported to be reduced compared to controls in diseases such as cystic fibrosis [5], Crohn's disease [6], ulcerative colitis [7], and colorectal cancer [8], suggesting that it is replaced or outcompeted in certain disease states. *E. rectale* is an important gut anaerobe, and it is thus crucial to study its population genetics and strain-level epidemiology.

The population structure of *E. rectale* has been investigated in previous studies [9–11], which have used read-mapping-based approaches to study the population-level genetics of bacterial commensals from metagenomes. Although these approaches provided valuable insights such as the variable degree of intra-species biogeographic stratification in different species including *E. rectale*, they were not conducted at the resolution of whole genomes. Metagenomic assembly together with reference-free binning has recently been employed in meta-analyses showing that microbial genomes can be consistently reconstructed from metagenomes [2, 12, 13]. However these reference-free binning approaches could miss genomic regions with divergent tetranucleotide frequencies.

In this work, we extracted more than 1300 high-quality *E. rectale* genomes from more than 6500 gut metagenomic assemblies using a targeted, reference-based binning approach that is applicable when at least a few (isolate) genomes are available. This pipeline produced genomes that compare favorably to genomes from a reference-free binning approach. The genomes that were assembled from metagenomes were used for the first large-scale genome-based population-level genomic analysis of *E. rectale* exemplifying how studies typically performed with cultured isolate sequencing data can be performed on carefully quality-controlled genomes from metagenomes. We extended the number of subspecies identified in previous investigations [9–11] by identifying a subspecies predominantly found in African individuals. Comparing median genetic distances to estimated geographic distances between pairs of subspecies indicated that pairs of subspecies are isolated by distance, in turn suggesting host-microbe co-dispersal. Whole-genome functional analysis confirmed the presence of a uniquely non-motile subspecies exhibiting loss of motility associated with a shift in carbohydrate metabolism gene repertoire.

## Results and discussion

### Reconstruction of > 1300 high-quality *Eubacterium rectale* genomes from > 6500 metagenomes

Metagenomic samples are a rich source for microbial genomes, but reconstructing bacterial genomes from metagenomes with sufficient completeness and accuracy remains challenging. To extract *E. rectale* genomes from metagenomes, we developed a three-step procedure consisting of (i) single-sample metagenomic assembly, (ii) compilation of high-quality *E. rectale* reference sequences, and (iii) use of these references to bin the metagenomic assemblies (Additional file 1: Fig. S1, "Materials and methods"). We applied this pipeline on a collection of 6775 gut metagenomic assemblies obtained from our previous studies [2, 14]. These assemblies were generated using metaSPAdes [15] if paired-end reads were available or MegaHIT otherwise [16]. We produced 47 manually curated reference (MCR) *E. rectale* genomes from 170 assembled metagenomes of diverse geographic origin in which *E. rectale* was particularly highly abundant ("Materials and methods"). These genomes are smaller than genomes obtained from isolate sequencing due to prioritization of specificity over sensitivity in the manually curated binning process ("Materials and methods"). For the last step of the pipeline, we used the 47 MCR genomes (Additional file 2: Table S1) together with seven *E. rectale* isolate genomes from NCBI available at the time (Additional file 3: Table S2) as references for the reference-based binning that was applied to all 6775 assembled metagenomes ("Materials and methods"). Semi-simulated metagenomic assemblies ("Materials and methods") allowed us to set optimal parameter values for the binning procedure (Fig. 1a).

We found that this pipeline reconstructs *E. rectale* genomes with high fidelity, outperforming reference-free metagenomic binning in terms of completeness [2, 18] while slightly increasing contamination (1.7% median increase in completeness, 0.5% median increase in contamination) (Fig. 1d, Fig. 1e). The pan-genome characteristics of the reconstructed *E. rectale* genomes more closely resemble those of isolate *E. rectale* genomes than the *E. rectale* genomes coming from reference-free binning (Fig. 1f, g), further suggesting that they generally are of high quality.

We obtained a total of 1321 high-quality (HQ) *E. rectale* genomes by applying our pipeline to a set of 6613 publicly available gut metagenomic assemblies as well as 162 gut metagenomic assemblies from rural populations in Madagascar and Ethiopia we recently sequenced [2, 19] (Fig. 1b, Additional file 4: Table S3). The combined cohort of 6775 gut samples encompasses 38 datasets from 30 countries with samples collected from individuals ranging in age from infants to elderly, and spanning different health conditions and lifestyles (Additional file 5: Table S4). The 1321 HQ *E. rectale* genomes contain less than 400 contigs and passed recently proposed completeness and contamination cutoffs (90% and 5% respectively) for high-quality metagenome-assembled genomes [20]. In line with recent large-scale metagenomic assembly efforts [2, 12, 13], we did not consider the presence of tRNA and rRNA genes as criteria for high-quality metagenome-assembled genomes because of the inherent difficulty of reconstructing genes that are conserved across related species [20]. The genomes were however further required to pass an additional quality measure we developed based on polymorphic site rates across core genes to flag genomes that are likely to incorporate strain-level variation from more than one strain ("Materials and methods"). The HQ genomes had an average length of 3.39 M bases (s.d. 0.22 M) and an average GC content of 41.47% (s.d. 0.27%), which was consistent with the genomes from isolate

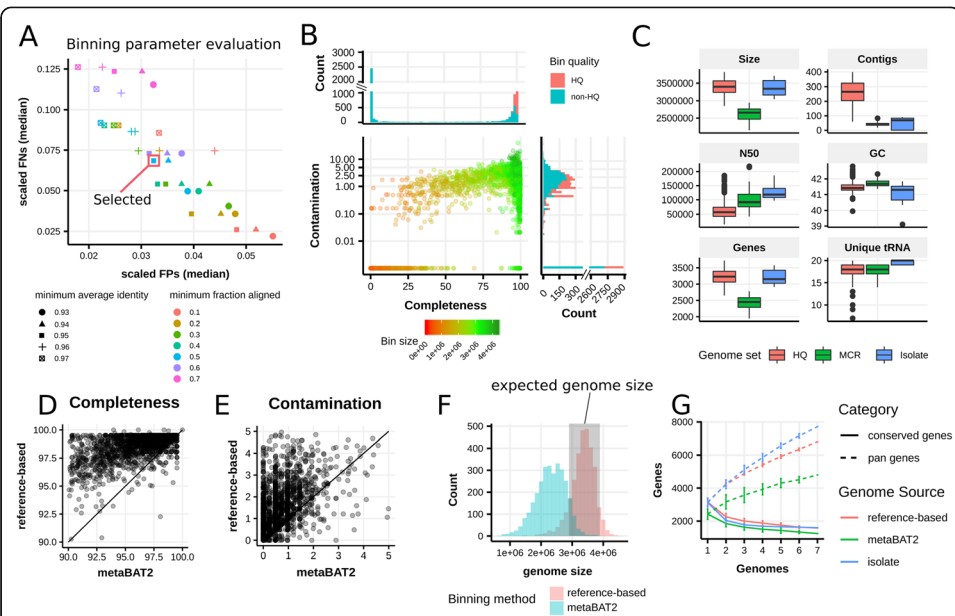

**Fig. 1** Reconstruction of 1321 high-quality (HQ) *E. rectale* genomes from 6775 fecal metagenomes. **a** The parameters for the binning step of our reference-based workflow (average identity and fraction of contig aligned) were chosen using *E. rectale*-free metagenomic assemblies spiked with *E. rectale* sequences obtained from isolate genomes ("Materials and methods"). We report the median number of false positive (FP) bases (binned contigs not coming from spike-in) and false negative (FN) bases (contigs coming from spike-in that were not binned). FP and FN values are scaled with respect to the average *E. rectale* isolate genome size. The red square indicates the parameter value combination used in this study. **b** Estimation of completeness and contamination for all extracted genomes using CheckM [17]. **c** Comparison of genome characteristics for *E. rectale* isolate genomes, genomes from metagenomes reconstructed with a semi-supervised approach (MCR), and the large set of automatically reconstructed genomes (HQ). **d, e** Completeness and contamination estimates for bins extracted using the reference-based binning approach used in this study and bins produced by a reference-independent pipeline using metaBAT2 [2, 18]. Only genomes with > 90% completeness and < 5% contamination in both approaches are shown. **f** The sizes of the *E. rectale* genomes reconstructed with the reference-based pipeline are very consistent with the genome sizes (gray area) from cultured isolate sequencing (gray shading) while the reference-independent pipeline produces genomes of smaller size. **g** Pan-genome characteristics for seven *E. rectale* isolate genomes from NCBI available at the time of processing (Additional file 3: Table S2) as well as seven genomes from the reference-based binning and from Pasolli et al. [2]. For both binning methods, we considered the same seven, randomly selected European metagenomes as well as all seven cultured isolate genomes originating from studies in Europe/North America

sequencing available for this species (Fig. 1c). The quality, number, and diverse nature of this combined cohort enabled us to undertake a large-scale genomic investigation of this currently under-characterized gut anaerobe species.

## A large-scale phylogeny refines *E. rectale* population structure and association with geography

To get an overview of the *E. rectale* population structure, we first performed a phylogenetic analysis of the 1321 HQ genomes in combination with eight publicly available cultured isolate genomes and two additional *E. rectale* isolates we sequenced for this work ("Materials and methods", Additional file 3: Table S2). The core gene concatenation approach we used ("Materials and methods") yielded 1071 core genes and a total alignment length of 1.02 M nucleotides. The maximum likelihood phylogeny and the ordination based on this alignment (Fig. 2a, b) confirmed previous observations that *E.*

*rectale* strains fall into discrete groups [9–11]. Clustering of core gene genetic distances using partitioning around medoids (PAM) [21] supported the existence of four subspecies (prediction strength consistently over 0.8 for $k = 4$, Additional file 1: Fig. S2, Fig. 2d, "Materials and methods"), one of which was not observed before [9–11]. Three of these four subspecies are large and well-defined monophyletic subtrees in the phylogeny, and only a minority of strains of the four *E. rectale* subspecies showed very strong geographic enrichment and were named accordingly. The three most represented subspecies correspond to what we designated as ErEurasia, ErEurope, and ErAsia as they predominantly comprised strains from these regions. ErAfrica, the fourth and previously unobserved subspecies, included strains derived mostly from sub-Saharan African countries (Madagascar, Tanzania, Liberia) but also contains strains from Peru and Indonesia (Fig. 2a, Fig. 2d). While ErAfrica, ErEurope and ErAsia are geographically relatively well contained, ErEurasia appears

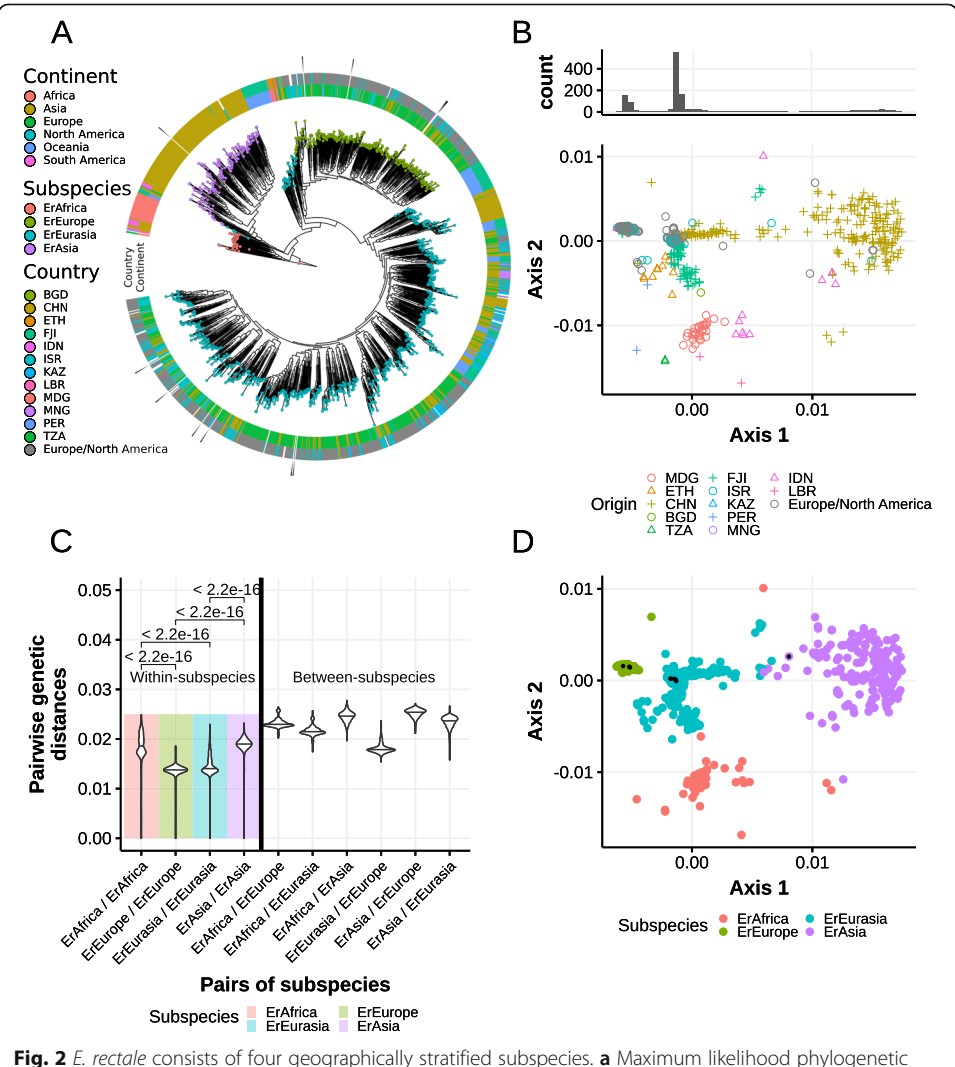

**Fig. 2** *E. rectale* consists of four geographically stratified subspecies. **a** Maximum likelihood phylogenetic tree of all *E. rectale* genomes, built from a concatenated core gene alignment using PhyloPhlAn2 ("Materials and methods") and rooted based on a phylogenetic tree including *E. rectale* sister species. **b** Non-metric multidimensional scaling plot of pairwise genetic distances between all *E. rectale* genomes. **c** Distribution of intra- and inter-subspecies core gene genetic distances. *p* values were obtained using bidirectional Wilcoxon rank-sum tests. **d** Subspecies assignment using PAM clustering with $k = 4$ ("Materials and methods"). Black points indicate genomes obtained from cultured isolate sequencing

to be comparatively widespread (Additional file 1: Fig. S3) with strains retrieved from gut metagenomes in Ethiopia and Fiji also belonging to this subspecies, albeit with divergent genetic makeup (Fig. 2b, Additional file 1: Fig. S4, Additional file 1: Fig. S5). Nonetheless, ErEurasia appears specifically enriched in central/northern Asian countries, with individuals from Kazakhstan, Mongolia, and Russia almost exclusively harboring genetically representative ErEurasia strains (Fig. 3a, Additional file 1: Fig. S4, Additional file 1: Fig. S6). While subspecies-specific SNV analysis confirmed that ErEurope and ErEurasia occasionally co-exist, the other subspecies almost never co-colonize (Additional file 1: Fig. S7, Additional file 1: Fig. S8, "Materilas and methods") and thus the geographic distribution inferred from our reconstructed *E. rectale* genomes does not obscure lowly abundant strains. Subspecies membership of the ten strains for which we had isolate genomes is congruent with their putative geographic origin (Additional file 3: Table S2), and while discrepancies between subspecies assignment and the geographical association of some strains exist (Additional file 1: Fig. S9), our data strengthens the notion of geographic stratification in *E. rectale* and provides a first approximation of the population structure of *E. rectale* on a global scale.

Genetic divergence between subspecies confirmed that they should be considered part of the same species as their pairwise genetic dissimilarities are well below 5%, which is the threshold typically used to define bacterial species [2, 22]. Indeed, the two most divergent subspecies are ErAsia and ErEurope which are at ~ 2.5% median genetic distance and no pair of strains ever exceeds 3% genetic distance (Fig. 2c, Additional file 1: Fig. S10). Nonetheless, the four subspecies have different intra- and inter-clade genetic variability. Strains belonging to ErEurope and ErEurasia have smaller intra-subspecies genetic variability (1.38% and 1.4% median variability, respectively) compared to ErAsia and ErAfrica (median 1.9% and 1.95%, respectively, Fig. 2c). ErEurope and ErEurasia are both the individually least genetically diverse and most closely related pair of subspecies.

*E. rectale* consists of at least four geographically stratified subspecies, exhibits differential within- and between-subspecies genetic variability (Fig. 2c), and is found in almost all adult control samples regardless of origin and lifestyle conditions (comprising differential levels of urbanization and sanitation as well as different

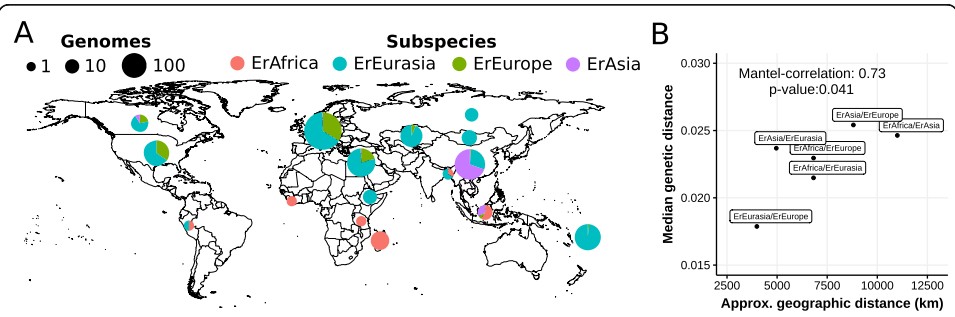

**Fig. 3** *Eubacterium rectale* subspecies distribution suggests subspecies are isolated by distance. **a** Relative prevalence of *E. rectale* subspecies per country (European countries are aggregated). The size of the pie charts is proportional to the total number of genomes obtained per region/country. For a map of Europe, see Additional file 1: Fig. S13. **b** Pairwise approximated geographic distances between subspecies (considering representative locations) correlate with their median genetic distances ("Materials and methods" for details). A Mantel test between pairwise genetic and geographic distances using the Pearson correlation coefficient yielded a correlation of 0.73 and a *p* value of 0.041

diets) (Additional file 1: Fig. S11, "Materials and methods"). Altogether, this showed that *Eubacterium rectale* is a globally spread human gut commensal and that the population genetics of *E. rectale* should be studied in light of the evolutionary relationship to its host.

### Correlation between subspecies' geographic and genetic distances suggests isolation by distance

An important aspect in investigating the evolutionary relationship between a microbe and its host is the level of host specificity and its transmission patterns. We screened for the presence of *E. rectale* in 146 publicly available metagenomes from wild non-hominid primates as well as 29 metagenomes from wild non-human hominids ("Materials and methods"). We found no evidence for the presence of *E. rectale* in any of these metagenomes using MetaPhlAn2 ("Materials and methods"). Similarly, none of the genomes assembled from non-human metagenomes is closely related (i.e., within 5% genetic distance) to any of the available *E. rectale* genomes. To assess the possibility of interindividual *E. rectale* strain transmission in human populations, we further analyzed metagenomic data from mother-infant pairs in multiple cohorts ($N = 532$ samples; "Materials and methods") and found evidence of vertical transmission (25% transmission rate within the first year of the infant's life, Additional file 1: Fig. S12). Overall, these analyses suggest that *E. rectale* is specific to humans and that it can be transmitted within populations.

Considering the reported specificity of *E. rectale* to humans, the differential degree of relatedness of *E. rectale* subspecies might be due to the effects of isolation by distance [23] and we thus tested whether *E. rectale* genetics supports this hypothesis. To this end, we compared median pairwise genetic distances with geographic distances between pairs of subspecies [24]. Owing to sparse sampling outside Europe and the occurrence of ErEurasia and ErAfrica strains outside their ascribed geographic areas, we assigned representative point locations to each subspecies that do not take these outlying strains into account ("Discussion") ("Materials and methods", Additional file 1: Fig. S14). Under these approximations, we found a statistically significant correlation (*p* value 0.041) between pairwise geographic and median genetic distances of subspecies (Fig. 3b) that is confirmed when directly considering pairwise distances between samples (*p* value <1e−16), suggesting that *E. rectale* genetic stratification could have been to some extent shaped by physical isolation of strains over time.

### ErEurope strains are immotile due to loss of motility operons

To analyze the functional repertoires of the *E. rectale* subspecies, we compared the presence and absence of functionally annotated gene clusters across all *E. rectale* genomes. ErEurope was found to be much more functionally divergent from the other subspecies than what genetic data would suggest (Additional file 1: Fig. S15). We computed differentially prevalent gene families (KEGG Orthology gene families, KOs) and found that the most distinguishing feature of ErEurope genomes is the absence of a large number of motility-related genes, some of which are part of an operon previously reported to be absent in a group of *E. rectale* strains corresponding to what we called ErEurope [11, 25]. Our analysis confirmed that many motility-related genes in *E. rectale* and in closely related species are organized in four operons [25] and showed that

ErEurope strains consistently and specifically lack all genes of these four motility operons (Fig. 4a), whereas the remaining *E. rectale* subspecies largely possess these operons.

To support the hypothesis that these operons are necessary for motility, we performed in vitro motility characterization tests in anaerobic conditions on six cultured *E. rectale* strains, two of which were not described before (two ErEurope and four ErEurasia isolates, Additional file 3: Table S2) and showed that the absence of these motility operons renders *E. rectale* strains immotile in vitro with a microscopy-based assay of motility (Fig. 4d, "Materials and methods"). The vast majority of non-ErEurope strains possess the motility operons, although there are a few exceptions (verified with contig-based analysis, Additional file 1: Fig. S16, "Materials and methods") with 41 non-ErEurope genomes (3.1% of all non-ErEurope strains) lacking > 20% of these motility genes (Fig. 4a, Additional file 1: Fig. S17) and 16 non-ErEurope genomes (1.2%, Additional file 1: Fig. S18) specifically missing the largest of the four operons (*flgB/fliA*, Fig. 4a). Within non-ErEurope strains, the genetic distances inferred from the *flgB/fliA* operon are highly correlated with those from the core genome (Pearson correlation of 0.8, Mantel test *p* value < 0.001, Fig. 4c, Additional file 1: Fig. S19). This suggests past operon/core genome co-diversification and thus that the common ancestor of all *E.*

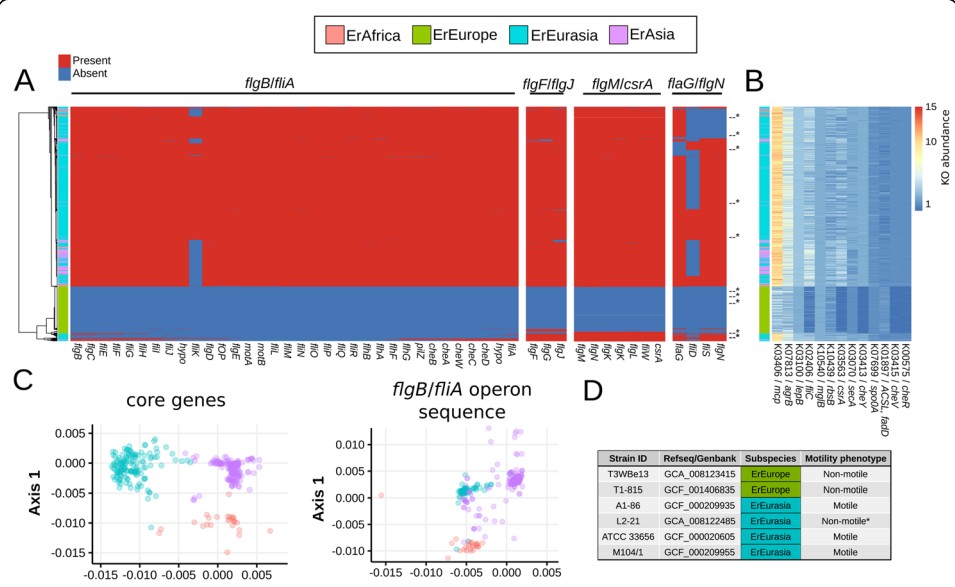

**Fig. 4** ErEurope is consistently immotile due to loss of motility operons. **a** No genes from the four motility operons of *E. rectale* [25] are detected in ErEurope strains, and only a very small fraction of non-ErEurope genomes are lacking some or all of these genes (Additional file 1: Fig. S18). Asterisks denote cultured isolate genomes. **b** Differentially abundant, non-operon potentially motility-associated KOs between ErEurope and the remaining subspecies. *csrA* was added despite being present in the *flgM/csrA* operon because it can be found elsewhere in some *E. rectale* genomes as well. We annotated genes using eggNOG-mapper [26] and only KOs of the *E. rectale* reference genome annotated by KEGG [27] are considered. Potentially motility-associated KOs were defined as being part of at least one of the following KEGG pathways: quorum sensing, bacterial chemotaxis, flagellar assembly, and two-component system. *p* values were calculated using a two-sided Wilcoxon test and corrected for multiple testing at 5% FDR using the Benjamini-Hochberg method. **c** Core gene sequence and *flgB/fliA* operon sequence genetic clustering for all motile strains (those belonging to either ErAfrica, ErEurasia or ErAsia). **d** In vitro motility characterization via phase-contrast microscopy of six *E. rectale* isolates ("Materials and methods"). Asterisk marks strain L2–21, which is the only immotile ErEurasia strain, presumably as a consequence of the specific lack of the *flgB/fliA* motility operon we found in its genomes

*rectale* strains possessed these operons. Motility operons show high structural consistency among *E. rectale* and closely related species, providing additional support for their homologous nature [25]. Together, we take this as evidence that operon motility loss in *E. rectale* is a stochastic event that can lead to viable, immotile *E. rectale* strains and that the subspeciation of ErEurope might be connected to one of such stochastic operon losses in the common ancestor of ErEurope strains largest subspecies is falling in divergent paraphyletic subtrees (Fig. 2a).

## Reduced genome size and increased functional divergence is associated with the loss of motility

Comparison of genome sizes between subspecies suggested that ErEurope strains have lost a considerable amount of genetic material since their split with ErEurasia, its most closely related subspecies. The median genome size of ErEurope is smaller than the median genome size of all other subspecies and 353 k bases smaller than that of ErEurasia (Additional file 1: Fig. S20). This difference far exceeds the cumulative length of the lost motility operons (mean cumulative size 54.5 kbps, sd 13 kbps., Additional file 1: Fig. S21), suggesting a gradual loss of genetic material.

We further investigated the evolutionary trajectories of subspecies by studying the differentiation of their gene repertoire in light of their genetic divergence. The gene distances between ErEurope and ErEurasia were similar to those between other pairs of subspecies (excluding motility operon genes) (Additional file 1: Fig. S22), but when normalized by their respective genetic distances, the resulting measure of the rate of functional divergence between ErEurope and ErEurasia strains clearly exceeded that of other pairs of subspecies (Additional file 1: Fig. S23), indicating that ErEurope and ErEurasia diverged functionally at an accelerated rate compared to other pairs of subspecies. This could represent adaptive processes triggered by the loss of motility in ErEurope strains.

## ErEurope genomes have reduced copy numbers of motility-associated genes that are not part of the four motility operons

We investigated the specific functions that are differentiating ErEurope and ErEurasia strains and found a total of 170 differentially abundant KEGG orthologous families (KOs) (Additional file 6: Table S5). Among them, we identified 13 KOs that were potentially motility-associated but were not found on any of the four motility operons (except for *csrA*, which can be found on a motility operon but also elsewhere in some genomes). Twelve of these 13 KOs are underrepresented in ErEurope strains (Fig. 4b).

The 12 out-of-operon, potentially motility-associated genes with reduced copy numbers in ErEurope comprised genes coding for proteins such as methyl-accepting chemotaxis protein (Mcp), flagellin (FliC), and the chemotaxis proteins CheR, CheY, and CheV (several other chemotaxis genes can be found on the *flgb/fliA* operon), which are directly involved in motility. This group also contained genes coding for proteins involved in cellular mechanisms that are indirectly related to motility, such as the accessory gene regulator B (*agrB*) that was shown to be involved in quorum sensing in *Staphylococcus aureus* [28], and the carbon storage regulator A (*csrA*) that is involved in biofilm formation in *E. coli* [29, 30] as well as quorum sensing in *Pseudomonas aeruginosa* [31]. The signal peptidase I (*lepB*) gene and the protein translocase subunit *secA* gene are both coding

for proteins required for protein export, a process crucial for flagellum assembly. We therefore speculate that the underrepresentation of these genes is the consequence of a gradual loss of these functionally redundant genes in early ErEurope strains.

### ErEurope strains have a distinct carbohydrate metabolism gene repertoire

To investigate whether carbohydrate metabolism gene repertoires differ between subspecies, we annotated all genomes using the CAZy database [32] ("Materials and methods"). We found that strains belonging to ErEurope harbor significantly more carbohydrate-active enzymes (all $p$ values < 1e−9, Fig. 5a) compared to the three remaining subspecies despite their smaller genome size. Consequently, ErEurope exhibits a much larger density of carbohydrate-active genes ($p$ value < 2.2e−16, Fig. 5b), and it clusters separately and distantly from the remaining subspecies also based on genome-wide CAZy gene content differences (Fig. 5c).

To understand in what way the carbohydrate metabolism of ErEurope strains has diverged from the other subspecies, we computed differentially abundant CAZy families between ErEurope and ErEurasia: in total, there were 43 differentially abundant CAZy families separating the two subspecies (Fig. 5d). ErEurope is enriched in putatively catabolic CAZy families (glycoside hydrolases, carbohydrate esterases, carbohydrate-binding module) targeting either hemicelluloses (xylans, arabinans, arabinoxylans) or pectins (galactans, arabinogalactans) (Fig. 5d). We performed in vitro carbohydrate utilization tests using six cultured *E. rectale* isolates (two of them belonging to ErEurope and four to ErEurasia) to understand on which carbohydrate substrates ErEurope strains grew better (optical density measured after 48 h of growth, Additional file 2: Table 1, "Materials and methods"). We found that, compared to ErEurasia strains, ErEurope strains grew better on xylan and inulin (both representing complex, plant-associated carbohydrates) and worse on sucrose. Furthermore, one of the two ErEurope strains was specifically able to grow on arabinan (Additional file 2: Table 1, "Materials and methods"). Together, these results indicate that ErEurope strains tend to be better at utilizing certain classes of complex, plant-associated carbohydrates compared to ErEurasia strains. These genomic differences might represent adaptive changes due to the loss of motility.

### A novel genomic island specific to ErEurope contains a battery of glycosyltransferase genes

Profiling the carbohydrate-related gene repertoire of *E. rectale* revealed another defining feature of ErEurope genomes. This subspecies is strongly enriched in genes coding for some glycosyltransferase (GT) gene families (Fig. 5d). Specifically, ErEurope strains possess more GT genes (from the families GT2, GT4 and GT32) compared to other subspecies, with GT2 being particularly strongly overrepresented ($p$ value < 1e−12, Fig. 6a). We found that the cultured *E. rectale* isolate genome T1–815 and several other ErEurope genomes derived from metagenomes contain a genomic region enriched in GT2, GT4, and GT32 genes (Fig. 6b) that is part of a genomic island (GI). This GI (when present) is consistently located in the same genomic position (Fig. 6c), and its GC content is clearly distinct from the remaining part of the genome (average GC content 37.7% vs 42.3%, Fig. 6d). The GI has a length of ~ 50 k bps (average 49,668 bps, s.d. 2176 bps) and is found in its entirety on the same contig in 56 ErEurope strains

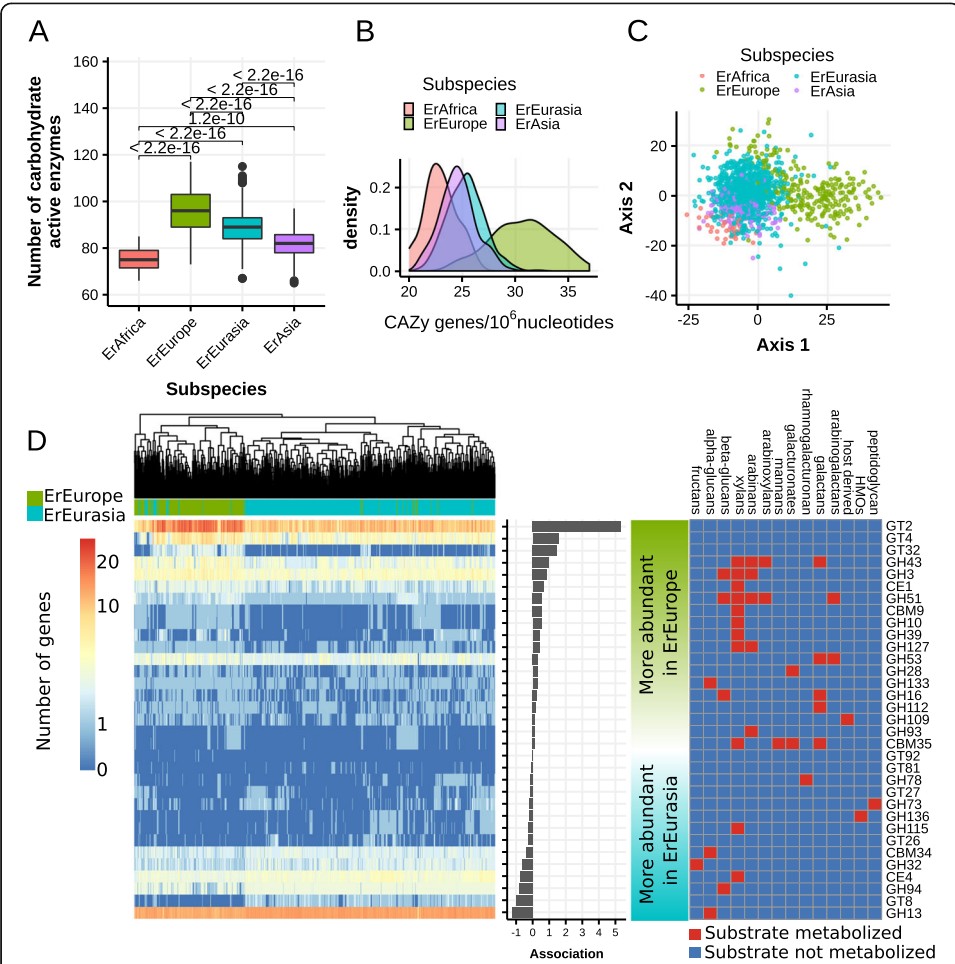

**Fig. 5** The immotile subspecies ErEurope exhibits a comparatively strong shift in carbohydrate-active enzyme (CAZy) gene repertoire. **a** ErEurope exhibits higher carbohydrate-active enzyme (CAZy) family counts than the other subspecies. **b** Density estimates of the number of CAZy genes per $10^6$ nucleotides in the genome for each subspecies. **c** Non-metric multidimensional scaling plot based on pairwise Manhattan distances between CAZy gene family abundances. **d** Left: Differentially abundant carbohydrate-active gene families between genomes of ErEurope and ErEurasia. *p* values were corrected at 5% family-wise error rate using the Bonferroni method. Color-scale is logarithmic. Middle: Effect size and direction of association (difference in mean copy number between ErEurope and ErEurasia). Right: Putative links between catabolic carbohydrate-active enzyme families (CBM, CE, GH) and their substrates. CBM = carbohydrate-binding module, CE = carbohydrate esterase, GH = glycoside hydrolase, GT = glycosyltransferase

(corresponding to ∼ 21% of all ErEurope genomes) with prevalence rates of up to 50% in ErEurope when partial detection of the GT-enriched region is considered sufficient to call the GI present (Additional file 1: Fig. S24). No traces of this GI were detected in any other subspecies.

The GT-enriched region of the GI is responsible for most of the enrichment of GT2, GT4, and GT32 gene families in ErEurope strains (Additional file 1: Fig. S25). While the non GT-enriched part of the GI remains largely functionally unannotated (Additional file 1: Table S6), many of the GT genes are associated with synthesizing exopolysaccharides in the context of biofilm formation, protein glycosylation, or cell wall polysaccharide synthesis. This may represent an adaptation of ErEurope strains to synthesize exopolysaccharides or other structural carbohydrates, a change that might prove advantageous for an immotile gut commensal.

**Table 1** In vitro carbohydrate growth assays ("Materials and methods"). The symbols represent growth (measured by OD at 650 nm after 48 h) as follows: "−": OD less than 0.1, "+": OD between 0.1 and 0.3, "++": OD between 0.3 and 0.7, "+++": OD greater than 0.7

| Subspecies | Strain | Negative control | Glucose | Raffinose | Sucrose | SPS | L-Arabinose | D-Arabinose | Inulin (chicory) | Inulin (dahlia) | Beta-glucan | Arabinan | Xylan |
|---|---|---|---|---|---|---|---|---|---|---|---|---|---|
| ErEurope | T3WBE13 | − | ++ | ++ | + | +++ | ++ | − | ++ | ++ | − | ++ | + |
| ErEurope | T1–815 | − | ++ | ++ | − | ++ | ++ | − | ++ | ++ | − | − | + |
| ErEurasia | A1–86 | − | + | ++ | ++ | +++ | ++ | − | + | + | − | − | + |
| ErEurasia | L2–21 | − | ++ | ++ | +++ | +++ | ++ | − | + | + | − | − | − |
| ErEurasia | ATCC 33656 | − | ++ | ++ | ++ | ++ | ++ | − | − | + | − | − | − |
| ErEurasia | M104/1 | − | ++ | ++ | ++ | +++ | ++ | − | + | + | − | − | − |

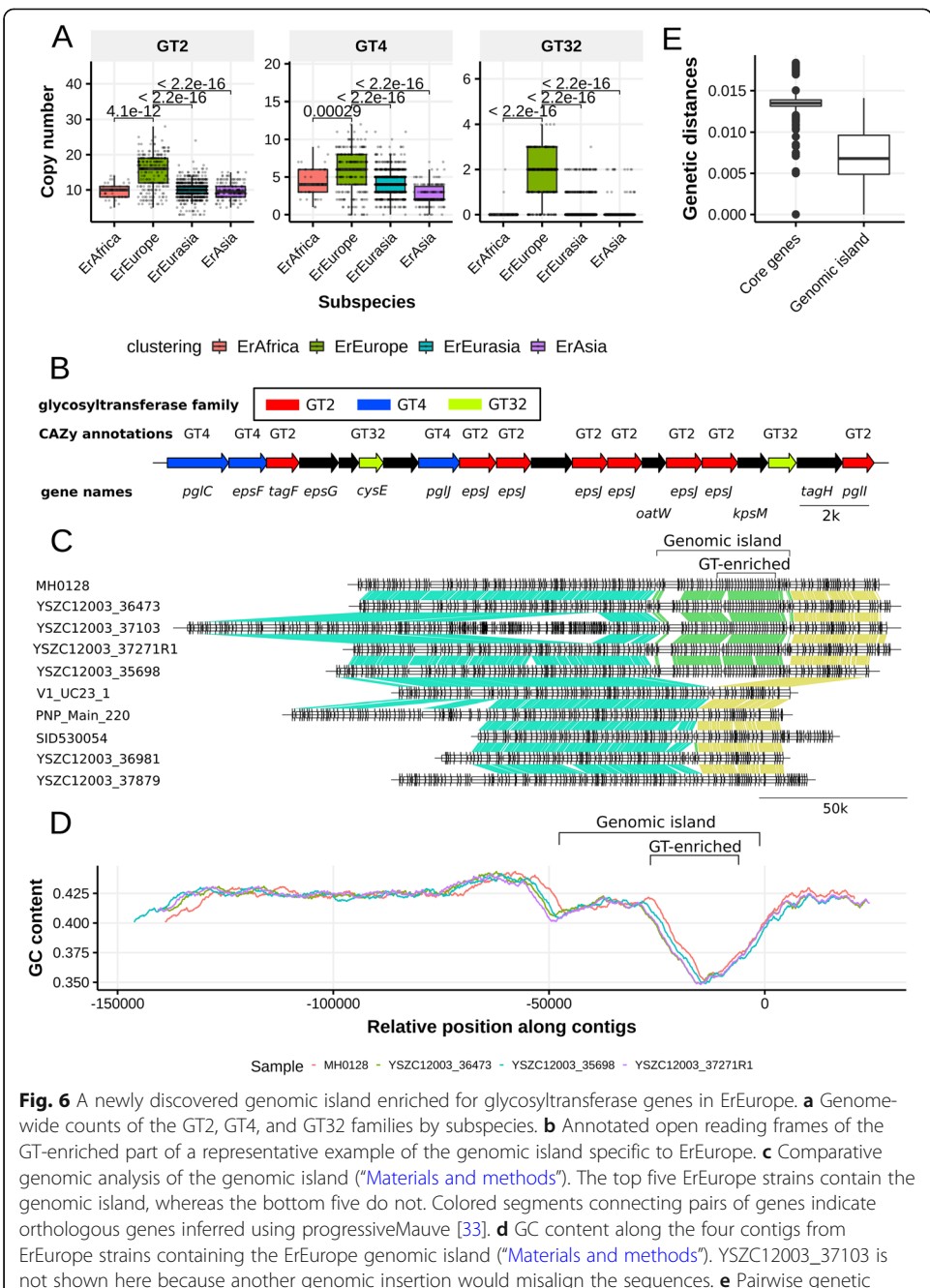

**Fig. 6** A newly discovered genomic island enriched for glycosyltransferase genes in ErEurope. **a** Genome-wide counts of the GT2, GT4, and GT32 families by subspecies. **b** Annotated open reading frames of the GT-enriched part of a representative example of the genomic island specific to ErEurope. **c** Comparative genomic analysis of the genomic island ("Materials and methods"). The top five ErEurope strains contain the genomic island, whereas the bottom five do not. Colored segments connecting pairs of genes indicate orthologous genes inferred using progressiveMauve [33]. **d** GC content along the four contigs from ErEurope strains containing the ErEurope genomic island ("Materials and methods"). YSZC12003_37103 is not shown here because another genomic insertion would misalign the sequences. **e** Pairwise genetic distances between strains using orthologous genes from the genomic island are lower than those based on core genes. All 56 ErEurope strains with fully extracted genomic island are considered here

In order to investigate the origin of the GI, we checked for signals of co-diversification between the core genome and operon sequences. The sequence of the GI is more conserved (< 1% pairwise genetic distance) than the rest of the core genome (Fig. 6e) and core gene distances and genomic island gene distances are significantly but very weakly correlated (Mantel test Pearson correlation 0.16, *p* value: 0.015). We screened the metagenomic assemblies of the human microbiome in search of homologous sequences of the GI, but found no evidence of any other human-associated microbe with the sequence of this GI ("Materials and methods"). This suggests that the

GI originated from a microbe which is not a common current member of the human gut microbiota.

## Discussion

New technologies and computational tools are generating an unprecedented amount of strain-specific genomic information that can be the foundation of a new generation of microbiome studies [2, 12, 13, 34, 35]. Large-scale species-specific whole-genome investigations can now be performed without cultivation [14] and—using metagenomic assembly combined with a reference-dependent binning approach—can be applied on many thousands of single metagenomes. We demonstrated this with *Eubacterium rectale*, one of the most prevalent human gut species.

Our analysis of *E. rectale* population structure revealed an extreme degree of biogeographic stratification and specificity to the human host. Our data largely supports the hypothesis that the observed stratification (Figs. 2b and 3a) is at least in part the consequence of isolation by distance (Fig. 3b) brought about by host-microbe co-dispersal, possibly due to migration movements of early humans. While population structure shaped by isolation by distance has previously been described for the (opportunistic) human pathogen *H. pylori* [36–38], here we report for the first time similar evolutionary signatures in a human gut commensal. Interestingly, vertical transmission rates were found to be low in both *H. pylori* [39, 40] and *E. rectale*. The estimated transmission rate of 25% observed between mother-infant pairs for *E. rectale* (Additional file 1: Fig. S12) suggests that strain seeding from the local (social) environment contributes to the observed biogeographic stratification.

However, isolation by distance is likely not the only force acting on the genetics of *E. rectale*. Most ErAfrica strains happen to originate from individuals living a traditional lifestyle. It is possible that selection effects by host lifestyle as is the case for *Prevotella copri* [14] influence the genetic structure of *E. rectale* strains as well. Since there are no large datasets that contrast individuals from the same population living different lifestyles, it is difficult to quantify the effect of host lifestyle on the population structure *E. rectale*. Nonetheless, we have tested for subspecies association with age (Additional file 1: Fig. S26) and BMI (Additional file 1: Fig. S27) as well as diet (Additional file 1: Fig. S28) and found no significant differences. Furthermore, ErAfrica strains are sometimes found in countries outside of Africa, and ErEurasia strains—despite being genetically distinct—are unexpectedly found in Fiji, observations that are not easily explained by isolation by distance. More comprehensive and better georeferenced metagenomic sampling of currently undersampled populations in South America, Africa, and Oceania that explicitly contrasts modern and traditional lifestyles will provide more conclusive answers. Powered by such data, our approach of large-scale genome reconstruction from metagenomes will open up new avenues to more broadly study the patterns of host-microbe co-evolution and co-differentiation.

*E. rectale* is consistently found in all cohorts used in this study and never found in wild non-human primates. This can suggest that the common ancestor of *E. rectale* was part of the gut microbiome of early humans prior to their expansion out of Africa. Bayesian phylogeny rooting did not support this hypothesis (Additional file 1: Fig. S29), but future studies exploring ancient DNA pools retrieved from prehistoric human gut content and sampling of undersampled populations (especially those from Africa) could shed further

light on the issue. Even without clocked phylogenies, key aspects in the genetic events that shaped the current dispersion of *E. rectale* strains could be found. Perhaps the most intriguing case is the evolutionary history of ErEurope that can be parsimoniously explained assuming a single operon loss event prior to its separation from early ErEurasia strains. This event must have occurred relatively recently compared to the other *E. rectale* subspeciation events as ErEurope has a comparatively low genetic diversity, is closely related to ErEurasia, and is geographically extremely well confined.

Studying the loss of major motility operons in ErEurope provided a detailed example of how large-scale strain-level metagenomics combined with experimental testing can reveal evolutionary and ecological patterns. ErEurope and its closest sister subspecies ErEurasia have functionally diverged at an accelerated pace after the loss of motility, and this is exemplified by the reduced number of extra-operon motility genes and the divergent carbohydrate metabolism gene repertoire in ErEurope. We speculate that, with the lack of motility, ErEurope strains might have been forced to change and extend their repertoire of catabolic carbohydrate-active enzymes to be able to metabolize a wider range of energetically unfavorable carbohydrates such as Inulin and Xylan (Additional file 2: Table 1) due to the inability to scavenge for energetically more favorable carbohydrates. A large genomic island specific to ErEurope was also identified that harbors many genes implied in exopolysaccharide synthesis in the context of biofilm synthesis. The loss of motility operons might have triggered a change in ecological niche in ErEurope strain, which in turn lead to adaptive processes in ErEurope with a combination of genome reduction and horizontal gene transfer events.

We provide an accurate, targeted approach to reconstruct genomes from metagenomes—based on a high-quality set of species-specific genomes (Additional file 1: Fig. S1)—which in our evaluation on *E. rectale* (Fig. 1d, Fig. 1e) compares favorably to a state-of-the-art reference-independent tool. The merit of such an approach needs to be further validated on other species and could then be useful for exploring diverse microbiomes including those from non-human environments. In the future, such efforts could be improved by technological advances including long-read technologies [41, 42] and single-cell sequencing [43], by even larger meta-analyses, and by in-depth phenotypic characterization that could pave the way to a deeper understanding of the complexity of the human microbiome on a subspecies level.

## Materials and methods

### Description of public and newly sequenced metagenomic datasets

We considered a total of 6775 human gut shotgun metagenomes from 38 datasets spanning 30 countries (Additional file 4: Table S3, Additional file 5: Table S4). Most samples were obtained from publically available datasets; a total of 163 samples came from new cohorts we recently sequenced: We included 113 samples from Madagasy individuals [2] and 50 samples from Ethiopian individuals [19]. The datasets we used are composed of individuals with different diets, exposure to environmental stressors (including antibiotics), and sanitary conditions. As such, some of those individuals can be described as "westernized" and others as "non-westernized" [44].

Furthermore, in this study we used four publically available datasets containing a total of 175 shotgun metagenomes coming from wild, non-human primates [45–48] (Additional file 8: Table S7).

### Prevalence testing of *E. rectale* in human and great ape metagenomes

#### Based on taxonomic profiling using MetaPhlAn2

All human- and non-human great ape samples were profiled using MetaPhlAn2 (version 2.7) [49] with default parameters. Reads were mapped to markers using Bowtie2 (version 2.3.4, parameters --very-sensitive, --no_unal) [50]. *E. rectale* was determined to be present in a sample if its relative abundance exceeded 0.1% and at least 20% of all *E. rectale* marker genes were hit.

#### Based on metagenomic assembly and binning

In order to find *E. rectale* genomes assembled from wild non-human primate metagenomes, we assembled and binned as described elsewhere [2, 51] a total of 2895 metagenomic high-quality genomes obtained from 175 publicly available metagenomes from wild, non-human primates (Additional file 8: Table S7). These 175 metagenomes come from four different datasets spanning 22 non-human primate species including chimpanzees and gorillas from 14 different countries on five continents [45–48]. We then estimated genetic distances between each of the reconstructed genomes and the set of *E. rectale* isolate genomes using MASH [52], and found that not a single bin generated from the non-human primates was within 23% genetic distance of any *E. rectale* isolate. To confirm that this result is not dependent on the binning method, we also applied the reference-based binning procedure we proposed in this work to these assemblies. We found that not a single bin was more than 5% complete, confirming our previous result that the metagenomic assemblies of wild non-human primates used in this study do not contain *E. rectale* genomes.

### Determining vertical transmission rates of *E. rectale*

Vertical transmission of *E. rectale* was assessed in three publicly available longitudinally sampled mother-infant datasets: Bäckhed et al. ($N = 398$ samples; 96 mothers-infant pairs) [53], Asnicar et al. ($N = 18$, 5 mother-infant pairs) [54], and Ferretti et al. ($N = 116$ samples, 21 mothers and 25 infants) [55]. Strain-level single-nucleotide variant profiling was performed with StrainPhlAn2 [9] with database version mpa_v294_CHOCOPhlAn_201901 and options sample_with_n_markers = 10 and marker_in_n_samples = 10. Pairwise genetic distances normalized by median branch length (nGD) were created using PyPhlAn (https://bitbucket.org/nsegata/pyphlan).

Strain transmission was assumed when two individuals harbored identical strains, with strain identity inferred when the pairwise normalize genetic distances are below the first percentile of the nGD distribution of samples of unrelated individuals, thus allowing a 1% false discovery rate. *E. rectale* transmission rates were defined as the proportion of mother-infant pairs harboring *E. rectale* that carried the same strain at ≥ 1 time point.

### The assembly and reference-based binning of *E. rectale* genomes from 6775 gut metagenomes

The reference-based binning approach employed here consists of three principal steps (Fig. 1a): Individual assembly of all 6775 gut metagenomes, compilation of a high-quality *E. rectale* genome set consisting of both isolate genomes and manually curated reference genomes from metagenomes and reference-based binning of all 6775 gut metagenomic assemblies using the high-quality *E. rectale* genome set as a reference for binning contigs.

First, we assembled each gut metagenome individually using metaSPAdes (version 3.10.1) with standard parameter settings [15] as described by Pasolli et al. [2]. We used MEGAHIT (version 1.1.1) [16] instead of metaSPAdes for those metagenomes with only unpaired reads.

Next, we compiled a set of *E. rectale* reference genomes consisting of manually curated metagenomic bins obtained using anvi'o (version 2.3.2) [56] as well as genomes from isolate sequencing. Anvi'o visually integrates information about depth, tetranucleotide frequency, and taxonomy of metagenomic assemblies on a contig-by-contig level, facilitating human-aided binning. We followed the author's recommended workflow for preparation of metagenomic assemblies for manual inspection ([http://meren-lab.org/2016/06/22/anvio-tutorial-v2/](http://meren-lab.org/2016/06/22/anvio-tutorial-v2/)). We complemented the taxonomic assignment provided by centrifuge (version 1.0.4) [57] with an ad hoc approach, mapping the assembled contigs against the bacterial RefSeq database using BLAST (version 2.6.0) [58]. Based on the results of this mapping, we assigned taxonomic labels to each contig of that species against which the largest fraction of the contig mapped with a mean identity score of at least 75%. Manually curated bins were used only when hierarchical clustering of tetranucleotide frequency and coverage as well as taxonomic assignments indicated an *E. rectale* bin of high quality. We maximized precision of manually curated reference genomes by excluding contigs that were not clearly belonging to *E. rectale*. A total of 170 metagenomes with high depth and coverage over *E. rectale* isolate genomes were queried in this manual binning process. From these 170 metagenomic assemblies, we reconstructed 47 manually curated reference genomes (MCR) with an average length of 2.61 Mbps (s.d. 0.20 Mbps), an average number of contigs of 41.1 (s.d. 11.14), an average N50 of 102,000 bps (s.d. 36,000 bps), and average CheckM [17] completeness and contamination estimates of 96.6% (s.d. 3.5%) and 0.2% (s.d. 0.3%), respectively (Additional file 2: Table S1). These MCR genomes have very good assembly characteristics (N50, nr. of contigs) but are shorter due to the maximization of precision during the manual curation step, which we expected to improve reference-based binning performance since the chance of faulty binning of small contigs from closely related species due to propagation of contamination in the reference is reduced.

The final step consisted of mapping all 6775 assembled metagenomes against the set of genomes consisting of the 47 manually curated reference *E. rectale* genomes (Additional file 2: Table S1) as well as seven isolate genomes from NCBI (Additional file 3: Table S2). We considered a contig to come from *E. rectale* if it mapped with a mean identity score of at least 95% over at least 50% of its length against the set of reference genomes. We determined optimal thresholds for minimum mean identity score/fraction mapping based on simulations with semi-synthetic data (see the section below).

The *E. rectale* metagenomic assemblies resulting from the procedure in this section were quality controlled and compared favorably against reference-free binning (see below).

### Parameter selection for the reference-based binning using semi-synthetic metagenomes

We used semi-synthetic data to select optimal parameter values in the reference-based binning approach. We spiked in sequences originating from *E. rectale* isolate genomes into metagenomic assemblies where *E. rectale* was undetectable using MetaPhlAn2 (version 2.7) [49]. We applied reference-based binning as outlined above and evaluated

performance over a grid of parameter values. The parameter values are (1) the mean identity score of the query contig against the reference database and (2) the fraction of the query contig mapping against the database. False positives are defined as those nucleotides binned that originated from the originally *E. rectale*-free metagenomic assembly; false negatives are defined as those spiked-in nucleotides that were not binned. The reference genomes which were not completely scaffolded were spiked in as they are, whereas the completely scaffolded reference genomes were sliced into uniformly distributed pieces between 1000 and 50,000 in length. We tested performance using all combinations of isolate genomes and 50 metagenomes without detectable levels of *E. rectale* (MetaPhlAn2) randomly chosen among all 6775 metagenomes.

### Comparison of reference-based against reference-free binning

We compared genomes extracted by the reference-based binning method described above with those from a large-scale, reference-free binning effort [2]. Briefly, the study by Pasolli et al. used metaBAT2 [59], a state-of-the-art reference-free binning software, on single-sample metagenomic assemblies to produce more than 150,000 genomes from metagenomes. The extracted genomes along with 80,990 reference genomes were clustered into species-level groups using pairwise genetic distances using MASH [52]. These groups were taxonomically labeled with the species associated with the reference genome(s) present in the group, considering the most common species label if multiple reference genomes with different assigned species were present. We selected the species-level group corresponding to *E. rectale* and compared those genomes that were more than 90% complete and less than 5% contaminated in both approaches. Very rarely, the reference-free binning by Pasolli et al. produced more than one bin assigned to *E. rectale* in a given metagenome. In these cases, only the more complete bin was evaluated. No longitudinal samples were considered.

### Quality control of the genomes

Filtering of genomes for downstream analysis consisted of removing lowly covered contigs in bins (those that are below 20% of the median genome-wide coverage) followed by further quality checks. High-quality (HQ) *E. rectale* genomes were defined as those with CheckM (version 1.0.12) [18] completeness > 90% and contamination < 5%, a total size larger than 2.9 Mbps and smaller than 3.89 Mbps (calculated as the 95% and 105% of the size of the smallest/largest *E. rectale* isolate genome), less than 400 contigs, and an estimate of within-sample strain heterogeneity of less than 0.3% (see below). As expected, the HQ genomes generally miss rRNA genes, containing on average 0.56 of them (sd 0.75). In total, we reconstructed 1321 HQ genomes that passed all these quality criteria and were used for further analysis.

### Polymorphism-based strain heterogeneity assessment for additional quality control

We developed a method for ad hoc estimation of within-metagenome strain heterogeneity for each genome based on the number of polymorphic sites over *E. rectale* protein-coding genes. After gene calling performed using Prodigal (version 2.6.3) [60], we mapped reads back to protein-coding genes using Bowtie2 (version 2.3.4, parameters --very-sensitive-local and -a) [50] and determined dominant and second-dominant

alleles over all protein coding nucleotides. For this, we only considered base calls with a PHRED quality score of at least 30 and only those positions with a coverage of at least 10. We considered a position non-variant if the dominant allele constituted more than 80% of the total number of nucleotides mapped to that given position. In order to calculate the polymorphism rate, we translated dominant and second-dominant nucleotide sequences into protein sequences and divided the total number of non-synonymous mutations between the two by the total number of positions.

### *E. rectale* genome annotation

We used Prokka (version 1.12) [61] for gene calling and functional annotation of bacterial genomes. Roary (version 3.8.2) [62] with settings "-i 95 -cd 95 -e -z --mafft" was used for core and pan-genome clustering as well as for generating core gene alignments [63]. Core genes were defined as those genes present in at least 95% of genomes. All gene clusters were annotated with KO information using eggNOG-mapper [26]) using representative gene sequences obtained from Roary. CAZy annotations [32] were obtained using a local dbcan distribution (release 6.0) [64], which uses HMMER (version 3.1b2) [65] to identify carbohydrate-active enzyme families in protein sequences. We used dbcan on translated protein-coding genes (Prodigal) and filtered hits for *E*-value < 1e−18 and coverage > 0.3 as suggested by the authors. Only one randomly selected sample per individual was considered for this analysis. Assignment of substrates to carbohydrate-active enzymes was based on the information provided in the CAZy database (www.cazy.org, [66]), CAZypedia (www.cazypedia.org, [67]), and dbCAN [64].

### Functional divergence rate

Genomic distances were calculated based on the Roary gene presence/absence matrix. Motility operon genes were identified by blasting representative operon gene sequences against representative gene sequences from roary and subsequently removed from the gene presence/absence matrix. Pairwise Jaccard distances between genomes were then computed using the "vegdist" function in the "vegan" R package. The genetic distances were defined as the hamming distance on a core gene alignment produced by roary. The rate of functional divergence was calculated by dividing pairwise inter-subspecies genomic distances by their corresponding genetic distance.

### Phylogenetic analyses

If not stated otherwise, the phylogenetic analyses were performed with PhyloPhlAn 3.0 [68] (https://github.com/biobakery/PhyloPhlAn).

The phylogeny in Fig. 2 was built using the 1071 core genes extracted as described above. PhyloPhlAn was run with the following options: "--diversity low --fast". For the internal steps, the following tools with their set of parameters were used:

- blastn (version 2.6.0+), [58] with parameters: "-outfmt 6 -max_target_seqs 1000000";
- mafft (version 7.310), [63] with the "--anysymbol" option;
- trimal (version 1.2rev59), [69] with the "-gappyout" option;
- RAxML (version 8.1.15), [70] with parameters: "-p 1989 -m GTRCAT".

To infer the Bayesian phylogeny, we built a core gene alignment (using an in-house script (https://bitbucket.org/CibioCM/genomealnbuilder)) from 46 metagenomes randomly selected to represent the four subspecies with the following parameters: "contigs_based -minqual 30 -minlen 50 -maxsnps 0.03 -mincov 5 -aln_len 500 -pid 95.0". We used trimAI to remove gappy columns from the alignment [69]. After filtering, the alignment included 1,356,039 nucleotide positions. BEAST v2.5.1 [71] was used to infer a phylogeny, using a GTR model of nucleotide substitution (with 4 gamma categories). To choose the best clock and demographic models we performed a model selection comparing coalescent constant, coalescent exponential, coalescent Bayesian skyline, and coalescent extended Bayesian skyline models (for the demographic priors) and a strict molecular clock. Convergence of the posterior probability distribution was assessed by visualizing log files with Tracer v1.7 [72]. The most fitting combination of models was a coalescent constant population with a strict molecular clock: this analysis was run longer for > 12,000,000 iterations with an effective sample size (ESS) of key parameters of over 200.

### E. rectale subspecies definition

To define subspecies, we used the partitioning around medoids algorithm [21] on the hamming distances (not considering gaps) computed on the concatenated nucleotide core gene alignment (produced by Roary, see above). In order to determine the optimal number of clusters, we used the prediction strength metric [73]. The PAM clustering algorithm is minimizing the sum of distances of each sample to the closest centroid, which is why it is prone to over-separate dense clouds of points. In order to produce more even sample densities, we subsampled all Eurasian/North American datasets to 50% and applied the PAM algorithm on this subset. We calculated prediction strength values on 50 random subsamples in order to obtain information regarding the variation of prediction strength values with respect to the subsamples. After having determined the optimal number of clusters ($k = 4$) following the standard procedure [73], we assigned cluster membership to all genomes based on the distance of each genome to the cluster corresponding to the closest centroid. We chose random cluster centroids from the 50 iterations, as the cluster assignment was very stable over subsamples for $k = 4$. We used the "pam" function in the "cluster" package and the "prediction.-strength" function of the "fpc" package in R with 20 internal divisions.

### E. rectale subspecies abundance estimation

We have used subspecies-specific single-nucleotide variants (SNVs) (defined using the core gene alignment as those nucleotides that are present in more than 90% of a subspecies but absent in more than 90% of the remaining ones) to estimate subspecies abundances in the samples. We mapped reads to consensus core gene sequences and—for each subspecies—calculated the median of the coverage ratios between the subspecies-specific alleles and the respective total coverages. We have restricted this analysis to only those metagenomes where the mean depth over all subspecies-specific positions was at least 5 and where at least 75% of the set of subspecies-specific positions was covered at least 3 times. We have removed samples where the sum of

estimated relative abundances is bigger than 1.25 or smaller than 0.75. For metagenomes passing these filters, we have scaled the estimated relative abundances to sum up to 1.

### Analysis of the *E. rectale* motility operons

The identification of the motility operons of *E. rectale* used in our analysis is based on the work by Neville et al. [25]. Briefly, Neville et al. determined and characterized motility operons in *E. rectale* and closely related species using isolate genomes. We annotated the operons in our genomes based on the *E. rectale* strain A1–86 used as a reference for the annotation by Neville et al. (Additional file 9: Table S8). Differently from the original analysis reporting the presence of three motility operons for *E. rectale*, we separated the *flgM/csrA* and *flaG/flgN* operons since we did not find them to be in close vicinity in both the genome of *E. rectale* strain A1–86 as well as in the genomes we reconstructed from metagenomes. In *E. rectale* strain A1–86, the largest operon (termed "*flgB/fliA*") has a total length of 30,520 nucleotides and 34 coding sequences. The three remaining operons have a length of 1984, 6764, and 4152 nucleotides and contain three, seven, and four coding sequences, respectively [25].

We determined the presence and absence of motility operon sequences using two different strategies. In the first, we extracted operon gene sequences from *E. rectale* strain A1–86 and blasted them against our genomes. We removed all hits that were shorter than 75% of query gene as well as redundant blast hits. After this, all hits had an identity score of at least 95% and all *E*-values were smaller than 1E–44 and were thus used to determine operon gene presence/absence. The second strategy involved extracting the genes immediately upstream and downstream of all motility operons (bordering gene sequences were taken from *E. rectale* strain A1–86), blasting them against all genomes and finding contigs on which both bordering genes of an operon could be found. We considered only those cases in which an operon could be identified well (exactly two blast hits per contig with an *E*-value < E–30). When a motility operon could be identified on a contig, we extracted all protein-coding genes between bordering genes and annotated them by mapping against the motility gene sequences of *E. rectale* strain A1–86.

### Analysis of exopolysaccharide genomic island
#### Detection of GT-enriched genetic element

We noticed a pronounced physical enrichment of genes annotated with glycosyltransferase activity in *E. rectale* isolate genome T1–815. We blasted this genetic stretch against all *E. rectale* genomes and extracted and aligned sequences (using mafft (version v7.310) [63] and standard parameter settings) in case there was a single blast hit with a length of at least 95% of the length of the genetic element of T1–815 (*E*-value cutoff of E–30), which was the case in a total of 56 ErEurope genomes. We further annotated the fully extracted sequences with Uniprot information [74] as provided by prokka (version 1.12).

#### Determining total size of genetic island

In order to determine the boundaries of this genetic island, we first blasted the GT-enriched sequence discovered in T1–815 against all *E. rectale* genomes and extracted exceptionally long contigs (contigs at least 100 k nucleotides long) with a single blast

hit of at least 30 k in size and an *E*-value of less than 1E−30. We then blasted these contigs back against all HQ *E. rectale* genomes, targeting strains of ErEurope with long contigs not enriched for GT genes since those represented strains without the genetic element. This two-step approach is necessary since the contig containing the genetic element from T1−815 is comparatively short (around 50 k bps) and was unable to attract contigs from ErEurope strains where the genetic element is absent. We aligned contigs with progressiveMauve [33] (build date Feb 132,015) and used only contigs that spanned the operon completely. When the genetic element borders were visualized, we used the first gene upstream/downstream of the genetic element that was inferred to be orthologous among all ten ErEurope genomes as the bordering genes. We calculated GC content along the contigs using a rolling window with window size of 20,000 and a step size of 10. Using this approach, each contig's 10 k positions to either end were not queried.

### Search for possible donor organism in gut metagenomic assemblies

In order to find a possible source microbe for the GT-enriched genomic island, we screened > 9500 human gut metagenomic assemblies [2] for a similar sequence. We blasted (blastn with parameters "-word_size 7") a representative sequence of the genomic island against all bins as well as all unbinned fractions of the metagenomic assemblies. Among the bins, we found several hits with a mean identity score > 98% across the entire length of genomic island in non-*E. rectale* bins. Yet, we noticed that in all those samples, a complete quality *E. rectale* genome was binned as well, which suggests that these contigs truly belong to *E. rectale*. We observed the same pattern in the unbinned fraction of contigs and concluded that this sequence might be unique among contemporary human gut commensals.

### Physical distances between subspecies

We estimated geographic distances between subspecies in order to look for a correlation between pairwise geographic and genetic distances of subspecies. We associated ErAfrica with Tanzania, ErEurope with Germany and ErAsia with Eastern China based on evident geographic enrichment (Fig. 3a). The geographic association of ErEurasia is less clear, with strains being found in Europe and central/northern Asia, but also in Fiji and Ethiopia, although strains from these two countries are genetically distinct from ErEurasia strains found in Eurasian countries (Fig. 3b, Additional file 1: Fig. S4, Additional file 1: Fig. S5). We associated ErEurasia with Kazakhstan because individuals from central/northern Asian countries (Kazakhstan, Mongolia, Russia) almost exclusively harbored genetically representative ErEurasia strains (Additional file 1: Fig. S30, Additional file 1: Fig. S31). Distances were approximated with the distm function of the geosphere package in R [75]. Physical distances between subspecies were defined as the shortest path between geographic locations associated with subspecies with the exception of the distance between ErAfrica and ErAsia, which was determined as the shortest path across the Arabian peninsula.

### Bacterial strains, isolation, and growth media

The bacterial strains L2−21 and T3WBe13 were isolated from human fecal samples from a healthy adult male consuming a vegetarian diet who had not taken any

antibiotics or other medication known to influence the human colonic microbiota for a period of more than 3 months prior to providing the samples. Strain L2–21 was isolated in 1995 as described previously [76]. T3WBe13 was isolated from another fecal sample from the same donor 22 years later. It was grown on clarified rumen fluid-based M2 medium [77] containing a range of soluble sugars (M2GSC containing glucose, cellobiose, and soluble starch, 0.2% final concentration of each) following a 10-fold serial dilution in basal M2 medium (containing 10% clarified rumen fluid) with 100 μl of slurry inoculated into 10-ml volumes of either M2 medium containing 0.2% wheat arabinoxylan (Megazyme) or 0.2% pre-treated bran as described previously [78]. Following 48 h incubation at 37 °C, the samples were enriched for a total of three times on the same medium prior to preparing a 10-fold serial dilution and inoculating roll tubes (M2GSC medium). Single colonies were picked into M2GSC broth. Strains A1–86, T1–815, ATCC 33656, and M104/1 have been described previously.

### Genome sequencing

The two genomes isolated and sequenced in this work (L2–21 and T3WBe13) were grown on M2GSC broths. Genomic DNA was extracted using the FastDNA SPIN Kit for Soil (MP Biomedicals). The sequencing libraries were prepared using the NexteraXT DNA Library Preparation Kit (Illumina, California, USA), following the manufacturer's guidelines. Library quality was assessed using the Caliper LabChip GX (high-throughput bioanalyzer) according to the manufacturer's instructions. The sequencing was performed on a HiSeq2500 machine (Illumina, California, USA).

### Experimental assessment of carbohydrate metabolism

The *E. rectale* strains were pre-grown overnight on M2GSC medium and inoculated into basal YCFA medium [79] containing individual carbohydrate substrates added at 0.2% w/v concentration. The carbohydrate sources tested were glucose (Sigma-Aldrich), Raffinose (Sigma-Aldrich), arabinan—sugar beet (Megazyme), soluble potato starch (Sigma-Aldrich), D-arabinose (Sigma-Aldrich), L-arabinose (Sigma-Aldrich), beta-glucan (Megazyme), inulin—chicory (Sigma-Aldrich), xylan—oat spelt (Sigma Aldrich), inulin—dahlia (Sigma-Aldrich), and sucrose (Fisher Scientific). As negative controls, cells were grown in basal YCFA with no added carbon sources. In total, 100 μL of each culture was then inoculated from its M2GSC growth medium into single carbohydrate or basal YCFA medium in triplicate under anaerobic conditions using oxygen-free $CO_2$ and incubated at 37 °C. Optical density measurements were taken spectrophotometrically after 48 h at a wavelength of 650 nm (Amersham Pharmacia Biotech, UK).

### Experimental validation of motility

In vitro screening for motility was tested using cultures grown to exponential phase (optical density 0.35–0.55) in M2GSC medium, then one drop was added to a dimpled glass slide anaerobically and covered with a glass coverslip. The wet mount was examined using phase-contrast to screen for motility. If individual cells were seen to be moving across the field of view, they were classified as motile.

## Supplementary information

---

**Additional file 1.** Supplementary Figures.

**Additional file 2.** Quality characteristics of 47 manually-curated reference (MCR) genomes used as references for the reference-based binning approach employed in this work to generate *E. rectale* genomes from metagenomes.

**Additional file 3.** Description of isolate genomes used in this study.

**Additional file 4.** Metadata for all samples used in this study.

**Additional file 5.** Summary of datasets used in this study.

**Additional file 6.** Differentially abundant KO groups between ErEurope and ErEurasia. A positive effect size means a given KO group is more abundant (on average) in ErEurasia. *P*-values were calculated using a two-sided Wilcoxon test. Q-values represent FDR-corrected (at 5%) *p*-values corrected using the Benjamini-Hochberg method.

**Additional file 7.** Annotations of gene calls in genomic islands found in ErEurope strains.

**Additional file 8.** Publically available datasets containing shotgun metagenomes from non-human primates used in this study.

**Additional file 9.** Mapping of motility operon genes and NCBI accessions based on Neville et al. [25]. The *flgM/csrA* operon coordinates/accessions could not be found in Neville et al., but could be inferred based on gene names provided by prokka. In contrast to what is stated in Neville et al., we could not find the *flgM/csrA* operon in proximity to the *flaG/flgN* operon in *E. rectale*.

**Additional file 10.** Review history.

---

### Peer review information

### Review history
The review history is available as Additional file 10.

### Authors' contributions
NS and NK conceived and supervised the study. EP, FeA, FrA, SM, PM, and MCC performed the data acquisition. NK, EP, AT, MVC, RR, ORS, MZ, DF, FrA, GZ, FM, CH, and KH performed the data analysis. DB, SHD, PL, and AW designed and performed the in vitro experiments. NK and NS performed the data interpretation and wrote the manuscript. All authors read and approved the final manuscript.

### Funding
This work was supported by NIH NHGRI grant R01HG005220, NIDDK grant R24DK110499, NIDDK grant U54DE023798, and CMIT grant 6935956 to C.H., and by the European Research Council (ERC-STG project MetaPG-716575), MIUR "Futuro in Ricerca" RBFR13EWWI_001, the European Union (H2020-SFS-2018-1 project MASTER-818368 and H2020-SC1-BHC project ONCOBIOME-825410), and the National Cancer Institute of the National Institutes of Health (1U01CA230551) to N.S. Further support was provided by the Programma Ricerca Budget prestazioni Eurac 2017 of the Province of Bolzano, Italy to F.M., and by the EU-H2020 (DiMeTrack-707345) to E.P. and N.S. D.B., S.H.D., P.L., A.W.W. and The Rowett Institute received core funding support from the Scottish Government Rural and Environmental Sciences and Analytical Services (SG-RESAS).

### Availability of data and materials
All datasets used in this study are publicly available and matched with their respective PMID (Additional file 5). The high-quality *E. rectale* MAGs in fasta format and a metadata file are available at http://segatalab.cibio.unitn.it/data/Erectale_Karcher_et_al.html and in the following Zenodo repository: https://doi.org/10.5281/zenodo.3763191 [80]. The two new isolate genomes L2–21 and T3BWe13 have been uploaded to NCBI and can be found in RefSeq under the accession numbers GCF_008122485.1 [81] and GCF_008123415.1 [82], respectively.

### Ethics approval and consent to participate
Not applicable.

### Consent for publication
Not applicable.

### Competing interests
The authors declare that they have no competing interests.

### Author details
[1]Department CIBIO, University of Trento, Trento, Italy. [2]Department of Agriculture, University of Naples, Naples, Italy. [3]Fondazione Edmund Mach, S. Michele all'Adige, Italy. [4]Rowett Institute, University of Aberdeen, Aberdeen, UK. [5]Free University of Bozen-Bolzano, Bolzano, Italy. [6]IATA-CSIC, Valencia, Spain. [7]EMBL, Heidelberg, Germany. [8]University of Bath, Bath, UK. [9]Institute for Mummy studies, Eurac Research, Bolzano, Italy. [10]Harvard T.H. Chan School of Public Health, Boston, MA, USA. [11]The Broad Institute, Cambridge, MA, USA.

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

## 

