## [**Additional file 10.** Review history. · Genome Biology]

Review History

First round of review

Reviewer 1

Are you able to assess all statistics in the manuscript, including the appropriateness of statistical tests used?

Yes, and I have assessed the statistics in my report.

Comments to author:

In their study, Karcher et al. conduct a large-scale study of the population structure of *E. rectale*, one of the most prevalent members of the gut microbiome. Their results largely confirm those presented in previous studies, namely that the *E. rectale* species contains genetically discrete subspecies that are geographically and functionally stratified, with one subspecies missing flagellar genes. While previous studies used read-mapping approaches, the current study used metagenome-assembled genomes. Additionally, inclusion of new data enabled the authors to identify one novel subspecies that predominates in Africans.

Major issues:

The claim in the abstract that the reconstructed genomes are "comparable in quality to those from the available cultured isolates" is not supported by the data, which indicates the opposite. Fig 1D clearly shows that the HQ MAGs are split across hundreds of contigs and have lower N50s compared to isolates. Related to this, it is also concerning that the MCHQ MAGs have a genome size that is ~1 Mb less than that of the isolates and HQ MAGs. Fig 1A indicates that MCHQ MAGs are 98% complete on average. How can these observations be reconciled? Did the authors throw away contigs from MCHQ MAGs that lacked a CheckM marker gene?

I found a number of issues with the hypothesis that the population structure of *E. rectale* is shaped by isolation by distance. First, other alternative hypotheses were not explored or considered. Namely, that the subspecies are globally distributed but selected for by environment/diet/lifestyle/age. Are subspecies patterns different within geographic regions between individuals living different lifestyles? Several datasets exist (including those analyzed by the authors) that could be used to evaluate this possibility. Second, I wonder how sensitive this result is to the choice of the number of subspecies. The main evidence for this hypothesis is a correlation of 0.73 (no p-value given) between the mean geographic and genetic distance between the 4 subspecies which is presented in Fig 3B. From looking at Figure 1A, there appears to be evidence of additional subspecies. For example, there appear to be two clearly distinct groups of genomes from Fiji that were assigned to the Eurasia subspecies. Third, the authors do not contextualize their results with the known routes of human migrations. How did an African subspecies end up in Peru? How did a European subspecies end up in Fiji? Fourth, the authors results are based on MAGs, which represent the dominant strain in each sample. I wonder if the observed distribution of subspecies would change if the authors used a read mapping approach. The authors could identify subspecies-specific genes, perform read mapping, and estimate the relative abundance of subspecies in each sample. I wonder if this would reveal that the subspecies are present at low abundances globally or not. Fifth, I wonder if strains of *E. rectale* are transmitted vertically or not. If *E. rectale* is shaped by human migration, then the expectation that is that it is passed from mother to infant. The data to answer this question (MAGs from mothers and infants) are likely included in the current study.

Minor issues:

The statement in the abstract regarding the loss of the motility operon confirms the results of a previous study (doi: 10.15252/msb.20177589). This should be indicated in the abstract to avoid the interpretation that this is a novel finding.

In the introduction, the authors characterize their study as "the first large-scale population-level genomic analysis of *E. rectale*". This is the first time MAGs have been used to study *E. rectale*, but not the first large-scale population-level genomic analysis.

Were the metagenomes assembled for the current study or assembled previously? This is not clear from lines 102-104.

On lines 113-114 the authors write that their pipeline greatly increased completeness at a small cost of contamination. From Fig 1E it looks like contamination was greatly increased. Please indicate summary statistics in the main text to support the above claim or reword as necessary.

The classification of high-quality MAGs (HQ) has been misappropriated. Beyond 90% complete and 5% contamination, the community-accepted definition requires a near-complete complement of tRNAs and rRNAs (doi: 10.1038/nbt.3893).

Please indicate what was the input to the PAM clustering algorithm. Was this % DNA identity across core genes between genomes? This is an important piece of information to report.

What was the rationale for using the PAM algorithm? From looking at their results in 2A, there are some subspecies assignments that disagree with the topology of the tree. Why did the authors not use the tree itself to define the lineages? Additionally, why the choice of 4 clusters (as opposed to 3 or 5)? I wonder how the conclusions in the paper would change with a different # of subspecies.

Reviewer 2

Are you able to assess all statistics in the manuscript, including the appropriateness of statistical tests used?
Yes, and I have assessed the statistics in my report.

Comments to author:

Overall this is an excellent analysis method applied to an important human commensal. The work uncovers some interesting novel biology and represents a significant advance that is of broad interest to the field.

The authors present a targeted approach to reconstruct genomes from metagenomes that is superior to "traditional" assembly based methods. Analysis of *Eubacterium rectale* genomes reconstructed from metagenomes suggests an interesting human specific, geographically distinct population structure with compelling differences in motility and carbohydrate utilisation. Technically the computational analysis is generally excellent with minor points of clarification required. The inclusion of experimental validations was positive but the manuscript would benefit substantially from greater coverage in both the figures and text of this data. Sufficient data is provided for replication of both informatic and experimental validation. The following points should also be addressed:

1. The authors have included 6775 human gut metagenomes (including newly sequenced samples to expand the diversity of donors; however, there are now at least 10,000 such human gut metagenomic samples available in ENA/NCBI on what basis were these samples selected and what impact may that have on the results described. This is particularly relevant for assessing the robustness of the newly proposed 3-step process and any biases that may exist with sample preparation, data quality, etc.
2. The phylogenetic tree (Figure 2a) appears to show members of the *ErEurasia* subspecies co-occurring with *ErAsia* and ancestral to *ErEurope* further comment should be made to explain this relationship given the apparent inconsistency between the genetic relatedness and the phylogenetic tree.
3. The authors state genomes assembled from non-human primates were not closely related to *E. rectale*, were these samples subjected to the same analysis performed on the human samples? This would provide the most compelling evidence and should be included and/or stated specifically.
4. Have the authors undertaken an assessment of the source studies from which the isolates originate? How many different studies contribute to each subspecies and could study related bias impact the observed results and/or apparent prevalence?
5. Given geographical separation is proposed as the driver of speciation the (relatively) limited number of functional differences identified is surprising. The specific order of analysis was unclear, were both presence/absence and sequence based selection considered for all genes? It may be of value to place additional focus on sequence based analysis in addition to differential abundance to determine if the inconsistencies observed within subspecies can be

explained by sequence variation, etc.

6. Could the authors speculate on the relevance/selective forces driving the loss of motility associated genes within this subspecies and the associations with carbohydrate metabolism? The rationale behind looking for carbohydrate metabolism

7. Experimental validation is an important and valuable addition to papers of this nature. I suggest further emphasis and analysis of the results should be included in the main text and figures for both the motility and carbohydrate utilisation.

Minor

1. The use of metaSPAdes with paired end and MegaHIT with unpaired reads could be mentioned in the main text for clarity.

2. Inclusion of terminology such as polymorphism based heterogeneity assessment on Line 129 rather than just referring readers to the methods would be preferable.

3. Separation of Figure 1 into two figures should be considered to improve clarity, at a minimum separation for Figure 1A.

4. The cultured isolates should be highlighted on the phylogenetic tree in Figure 2A

5. Colours on Figure 2A could be improved to more clearly separate colours for subspecies, continent and country. Also consistency of colours between sub-part of figures (e.g. 2A and 2B) would make the patterns clearer to the reader

6. The authors should carefully choose their wording regarding the association with an ancient link as the presented data and phylogenetic dating does not seem to prove this conclusively.

Point-by-point response to the reviewers

Reviewer #1

In their study, Karcher et al. conduct a large-scale study of the population structure of *E. rectale*, one of the most prevalent members of the gut microbiome. Their results largely confirm those presented in previous studies, namely that the *E. rectale* species contains genetically discrete subspecies that are geographically and functionally stratified, with one subspecies missing flagellar genes. While previous studies used read-mapping approaches, the current study used metagenome-assembled genomes. Additionally, inclusion of new data enabled the authors to identify one novel subspecies that predominates in Africans.

Major issues:

(1) The claim in the abstract that the reconstructed genomes are "comparable in quality to those from the available cultured isolates" is not supported by the data, which indicates the opposite. Fig 1D clearly shows that the HQ MAGs are split across hundreds of contigs and have lower N50s compared to isolates. Related to this, it is also concerning that the MCHQ MAGs have a genome size that is ~1 Mb less than that of the isolates and HQ MAGs. Fig 1A indicates that MCHQ MAGs are 98% complete on average. How can these observations be reconciled? Did the authors throw away contigs from MCHQ MAGs that lacked a CheckM marker gene?

We thank the reviewer for this remark and we agree we did not phrase this accurately. The genomes we reconstructed are on average indeed more fragmented than those from isolates and the word 'comparable' is rather subjective and context-dependent. We have now removed the statement on the comparison to isolate genomes in the abstract.

Abstract: Here, we leverage metagenomic assembly followed by a reference-based binning strategy to screen >6,500 gut metagenomes spanning geography and lifestyle and reconstruct >1,300 *E. rectale* high-quality genomes from metagenomes

The smaller genome size of MCHQ bins (compared to isolate genomes/high quality MAGs) is instead fully expected, but we agree we did not explain well why this is the case. We explicitly valued specificity over sensitivity in the manually-assisted binning process that generated the MCHQ genomes and this is the reason these genomes are smaller. We prioritized specificity because contamination in these references was expected to result in contamination propagating to the binned genomes (since our procedure is reference-based). We now explain the reduced genome size of MCHQ genomes in the main text and expanded on the rationale in the methods.

Main text: These genomes [MCHQ genomes] are smaller than genomes obtained from isolate sequencing due to prioritization of specificity over sensitivity in the manually-curated binning process (**Methods**).

Methods: These MCHQ genomes have very good assembly characteristics (N50, nr. of contigs) but are shorter due to the maximization of precision during the manual curation step, which we expected to improve reference-based binning performance since the chance of faulty binning of small contigs from closely related species due to propagation of contamination in the reference is reduced.

The reviewer also wondered how the small size of MCHQ genomes can be reconciled with their very high completeness estimates and wondered whether we excluded any contigs. We did not exclude contigs that lacked CheckM markers, because this would have removed a large portion of the accessory genome. The selection during manual curation of genomes from metagenomes encompasses clustering of contigs by tetra-nucleotide frequency and coverage (visualized in anvio) and probably resulted in bins missing contigs without checkM marker genes. However, this is not an issue because MCHQ genomes were only used in the reference-based binning (in combination with the larger genomes from isolate sequencing) and were not used as genomes for any subsequent analyses, and therefore their shorter size does not directly affect any results or conclusions on the comparative genomics.

To incorporate these modifications and clarifications in the revised version of the manuscript we also decided to rename the MCHQ genomes to Manually-curated Reference genomes (MCR), which better reflects their purpose.

I found a number of issues with the hypothesis that the population structure of *E. rectale* is shaped by isolation by distance.

We thank the reviewer for the constructive comments and we reply to each of them below. We would like to take the chance to emphasize that our intention is to propose a hypothesis, i.e. that isolation by distance can be one of the evolutionary forces shaping *E. rectale* population structure, rather than stating that *E. rectale* is solely shaped by isolation by distance. Our hypothesis is supported by the data that we now better contextualize and for which we now better discuss its limitations. This hypothesis is only one of the main points of the paper and it is clear that it will need additional work in the future. Also, it is important to mention again that our data indicates that isolation by distance is a major - as opposed to the only - evolutionary force that shaped the genetic structure of *E. rectale*.

First, other alternative hypotheses were not explored or considered. Namely, that the subspecies are globally distributed but selected for by environment/diet/lifestyle/age. Are subspecies patterns different within geographic regions between individuals living different lifestyles? Several datasets exist (including those analyzed by the authors) that could be used to evaluate this possibility.

It is true that there are other effectors that might affect the population structure of *E. rectale*. Lifestyle is indeed possibly a confounding factor: most ErAfrica strains happen to come from individuals living a more traditional lifestyle. However, we performed now additional analyses that point at a minor role of westernization in the current population structure of *E. rectale*. In particular, we (i) highlighted the presence of strains from typically westernized populations in non-westernized population (ii) we showed that *E. rectale* subspecies are not globally distributed, and most importantly (iii) we show that testable metadata (age, BMI, diet) mentioned by the reviewer are not associated with subspecies.

On (i), there are unfortunately no studies that sampled individuals from the same population living different lifestyles and we can thus not test this directly. Nonetheless, many strains in several non-westernized cohorts in very distinct geographic locations (i.e. from Fiji and from African populations) are assigned to different subspecies, suggesting that westernization cannot be the main factor influencing *E. rectale* population structure.

On (ii), we show in this revision that *E. rectale* strains belonging to different subspecies do not co-occur (see **answer to the fourth point**), except for ErEurope and ErEurasia, which are found in the same

populations in Europe and North America. This suggests that *E. rectale* subspecies are not all found everywhere (so not “globally distributed” as per reviewer’s question) and that the subspecies stratification we report here is a good approximation of true strain distributions.

We have expanded the results section accordingly as reported in the **answer to the fourth point**.

On (iii), regarding the association of other metadata, such as age/BMI/diet, with subspecies distribution, we performed additional analyses showing that age, BMI and diet do not appear to have a significant association with subspecies distribution in the datasets we checked (From Eurasian countries and China). For age and BMI, we had several datasets available, whereas for diet (vegetarian/non-vegetarian), we could use only two as unfortunately other studies that considered dietary data did not make these data publicly available. These new analyses, that we included as three Supplementary Figures reported below, showed that there are no significant differences in age, BMI or diet between subjects harbouring different subtypes after multiple testing correction. Only when multiple testing correction is not performed, one dataset (QinN_2014) shows a significant difference in age between people harbouring ErEurasia and ErAsia, but we attribute this to a false positive result due to the large number of tests and we note that this trend is inconsistent across the other datasets tested. It is thus unlikely that these variables are strongly associated with subspecies distribution in *E. rectale*.

Supplementary Figure 29: Boxplots of age grouped by subspecies. Label corresponds to significance level at 5% FDR (FDR-correction using Benjamini-Hochberg), numbers in parenthesis correspond to uncorrected p-values. P-values calculated using two-sided Wilcoxon tests.

Supplementary Figure 30: Boxplots of BMI grouped by subspecies. Label corresponds to significance level at 5% FDR (FDR-correction using Benjamini-Hochberg), numbers in parenthesis correspond to uncorrected p-values. P-values calculated using two-sided Wilcoxon tests.

Supplementary Figure 31: Bar Plots showing the distribution of ErEurope and ErEurasia in two datasets where qualitative diet information (Vegetarian/Non-Vegetarian) was available. P-values were calculated using a two-sided Fisher test.

Second, I wonder how sensitive this result is to the choice of the number of subspecies. The main evidence for this hypothesis is a correlation of 0.73 (no p-value given) between the mean geographic and genetic distance between the 4 subspecies which is presented in Fig 3B. From looking at Figure 1A, there appears to be evidence of additional subspecies. For example, there appear to be two clearly distinct groups of genomes from Fiji that were assigned to the Eurasia subspecies.

The correlation of 0.73 is statistically significant with a p-value of 0.041 as we originally reported in the caption in **Figure 3B**. We now include the p-value also in the Figure. However, the point of the reviewer is well taken, and so to ensure that the test for isolation by distance employed in this study is not dependent on the number of subspecies, we performed a subspecies-independent Mantel correlation test of geographic against genetic distances between all pairs of genomes in Eurasia and Africa (not including Fijian strains), which yielded a significant p-value ($<1E-16$) and a robust Pearson correlation coefficient of 0.55. Inclusion of the Fijian strains still yields a significant p-value ($<1E-16$), but the correlation coefficient is reduced to 0.23. It is true that some of the Fijian strains were assigned to ErEurasia (despite showing

more fine-grained genetic variation, **Fig. 1**), but we cannot explain the genetic stratification in the Fijian population using isolation by distance (See **answer to third point**). Nevertheless, this test shows that the choice of the number and assignment of subspecies does not impact the observed effect of isolation by distance. We now also mention the strain-wise isolation by distance test in the main text.

Main text: Under these approximations, we found a statistically significant correlation (p-value 0.041) between pairwise geographic and median genetic distances of subspecies (Fig. 3B) that is confirmed when considering directly pairwise distances between samples (p-value < 1e-16), suggesting that *E. rectale* genetic stratification could have been to some extent shaped by physical isolation of strains over time.

Although the number of subspecies does not impact the result on the isolation by distance, we agree it is still relevant to better discuss how this number was determined. This number was not manually or a-priori chosen, but instead determined using a cross-validation based assessment of cluster strength (prediction strength (Tibshirani and Walthers 2005)), which evaluates how consistent cluster assignments are after splitting data randomly into two folds, together with the Partitioning Around Medoids clustering algorithm on genetic distances between genomes based on core gene hamming distances. This is a validated and extensively used approach in the field to identify dense and well-separated clusters (Costea et al. 2017; Costea et al. 2018; Arumugam et al. 2011; Koren et al. 2013). This procedure is described in the Methods. We argue that no discrete clustering can recapitulate all genetic variation, which is a common problem of subspecies/subtype/strain definition in microbial genetics but we believe our assignment approach is rigorous, independent from metadata, consistent with the literature, and captures the largest degree of genetic stratification that can be consistently recapitulated over subsamples. It is important to emphasize that the determined, optimal number of subspecies is agnostic with respect to the testing of isolation by distance.

Third, the authors do not contextualize their results with the known routes of human migrations. How did an African subspecies end up in Peru? How did a European subspecies end up in Fiji?

As the reviewer correctly hints at, there is some genetic variation which cannot be explained by a model of host-microbe co-migration as we already mentioned and discussed above in this response. Yet, it is clear even in the well-established case of *H. pylori* (for which very extensive data has been generated that strongly and consistently supports a co-migratory link with humans) that not all sampled strains can be expected to be in accordance with a strict model of co-migration (Moodley 2016). It is possible that the strains in Peru and Fiji have been replaced in the process of recent migration/inter-population contacts or have admixed with other populations. In the case of *H. pylori*, the European subtype (HpEurope) was only very recently inferred to be the result of an admixture event between two ancestral populations (Moodley 2016). It is not possible to assess if the Fijian/Peruvian strains we report represent the subspecies predominating in native individuals of those geographic regions or whether they were indeed replaced recently and we sampled those strains by chance. Unfortunately, we do not have nearly enough sampling breath over South America, Africa, Australia and Oceania to answer this question. Furthermore, we reinforce in the discussion that isolation by distance is not the only force at action. Still, we believe the data suggests that isolation by distance likely played a prominent role in shaping *E. rectale*'s population structure and we believe it is worthwhile to present it.

We have changed and extended the results and discussion substantially to better articulate these important caveats.

Main text: Considering the reported specificity of *E. rectale* to humans, the differential degree of relatedness of *E. rectale* subspecies might be due to the effects of isolation by distance [23] and we thus tested whether *E. rectale* genetics supports this hypothesis. To this end, we compared median pairwise genetic distances with geographic distances between pairs of subspecies [24]. Owing to sparse sampling outside Europe and the occurrence of ErEurasia and ErAfrica strains outside their ascribed geographic areas, we assigned representative point locations to each subspecies that do not take these outlying strains into account (**Discussion**) (**Methods, Suppl. Fig. 15**). Under these approximations, we found a statistically significant correlation (p-value 0.041) between pairwise geographic and median genetic distances of subspecies (**Fig. 3B**) that is confirmed when considering directly pairwise distances between samples (p-value < 1e-16), suggesting that *E. rectale* genetic stratification could have been to some extent shaped by physical isolation of strains over time.

Discussion: Our analysis of *E. rectale* population structure revealed an extreme degree of biogeographic stratification and specificity to the human host. Our data largely supports the hypothesis that the observed stratification (**Fig. 2B, Fig. 3A**) is at least in part the consequence of isolation by distance (**Fig. 3B**) brought about by host-microbe co-dispersal, possibly due to migration movements of early humans. While population structure shaped by isolation by distance has previously been described for the (opportunistic) human pathogen *H. pylori* [36–38], here we report for the first time similar evolutionary signatures in a human gut commensal. Interestingly, vertical transmission rates were found to be low in both *H. pylori* [39,40] and *E. rectale*. The estimated transmission rate of 25% observed between mother-infant pairs for *E. rectale* (**Suppl. Fig. 13**) suggests that strain seeding from the local (social) environment contributes to the observed biogeographic stratification.

However, isolation by distance is likely not the only force acting on the genetics of *Eubacterium rectale*. Most ErAfrica strains happen to originate from individuals living a traditional lifestyle. It is possible that selection effects by host lifestyle as is the case for *P. copri* [14] influence the genetic structure of *E. rectale* strains as well. Since there are no large datasets that contrast individuals from the same population living different lifestyles, it is difficult to quantify the effect of host lifestyle on the population structure *E. rectale*. Nonetheless, we have tested for subspecies association with age (**Suppl. Fig. 30**) and BMI (**Suppl. Fig. 31**) as well as diet (**Suppl. Fig. 32**) and found no significant differences. Furthermore, ErAfrica strains are sometimes found in countries outside of Africa, and ErEurasia strains - despite being genetically distinct - are unexpectedly found in Fiji, observations that are not easily explained by isolation by distance. More comprehensive and better georeferenced metagenomic sampling of currently undersampled populations in South America, Africa and Oceania that explicitly contrasts modern and traditional lifestyles will provide more conclusive answers. Powered by such data, our approach of large-scale genome reconstruction from metagenomes will open up new avenues to more broadly study the patterns of host-microbe co-evolution and co-differentiation.

Fourth, the authors results are based on MAGs, which represent the dominant strain in each sample. I wonder if the observed distribution of subspecies would change if the authors used a read mapping approach. The authors could identify subspecies-specific genes, perform read mapping, and estimate the relative abundance of subspecies in each sample. I wonder if this would reveal that the subspecies are present at low abundances globally or not.

This is a very interesting point, and following up on this suggestion we performed analysis in this revision to identify lowly-abundant and non-dominant *E. rectale* subspecies from shotgun metagenomic reads.

Because subspecies-specific genes were not very suitable in this case (not enough subspecies-specific genes to obtain reliable results), we instead used subspecies-specific single-nucleotide variants (SNVs).

Subspecies-specific SNVs are defined on the core gene alignment as those nucleotides that are present in more than 90% of the strains in a subspecies but absent in more than 90% of the strains in all other subspecies. To estimate subspecies abundances in the samples we then mapped reads to consensus core gene sequences and - for each subspecies - calculated the median coverage ratio between the subspecies specific alleles and the total coverage for all subspecies-specific SNVs (**See Method text at the end of this paragraph**).

The results of this new analysis suggest that only ErEurope and ErEurasia are occasionally found to co-occur. The remaining subspecies appear to co-occur only in an extremely limited number of cases, possibly reflecting technical noise (**See Figures below**).

Since only ErEurope and ErEurasia were found to occasionally co-occur within European countries, and both ErEurope and ErEurasia MAGs were already assigned to European countries based on the dominant strain, the SNV-based analysis confirms the geographic distribution we inferred from our MAGs.

We have added Supplementary Figures, method text and explain this result in the main text.

Main text: While subspecies-specific SNV analysis confirmed that ErEurope and ErEurasia occasionally co-exist, the other subspecies almost never co-colonize (**Suppl. Fig. 8, Suppl. Fig. 9, Methods**) and thus the geographic distribution inferred from our reconstructed *E. rectale* genomes does not obscure lowly abundant strains.

Methods:

E. rectale subspecies abundance estimation

We have used subspecies-specific Single Nucleotide Variants (SNVs) (defined using the core gene alignment as those nucleotides that are present in more than 90% of a subspecies but absent in more than 90% of the remaining ones) to estimate subspecies abundances in the samples. We mapped reads to consensus core gene sequences and - for each subspecies - calculated the median of the coverage ratios between the subspecies specific alleles and the respective total coverages. We have restricted this analysis to only those metagenomes where the mean depth over all subspecies-specific positions was at least 5 and where at least 75% of the set of subspecies-specific positions was covered at least 3 times. We have removed samples where the sum of estimated relative abundances is bigger than 1.25 or smaller than 0.75. For metagenomes passing these filters, we have scaled the estimated relative abundances to sum up to 1.

Supplementary Figure 8: Barplots of subspecies relative abundances over all metagenomic samples that had sufficient coverage over subspecies-specific SNVs (**Methods**).

Supplementary Figure 9: Boxplots of subspecies relative abundances over all metagenomic samples that had sufficient coverage over subspecies-specific SNVs grouped by the dominant subspecies in the sample.

Fifth, I wonder if strains of *E. rectale* are transmitted vertically or not. If *E. rectale* is shaped by human migration, then the expectation is that it is passed from mother to infant. The data to answer this question (MAGs from mothers and infants) are likely included in the current study.

According to data encompassing three datasets with mother-infant pairs (Baeckhed_2015, Asnicar_2017, Feretti_2018), 25% of infants carrying *E. rectale* carry the same strain as their mother, suggesting that *E. rectale* can indeed be vertically transmitted. These three mother-infant studies are limited to a short timespan, do not consider non-dominant strains, and do not consider other family members, so the 25% vertical transmission reflects a lower bound on the transmission rate. So, while we showed that *E. rectale* can be vertically transmitted from mother to infant, it is likely that it is transmitted vertically also from other family members (including later on in life) as well as horizontally from other members of the population (which are in close physical proximity).

For *H. pylori*, a very well-understood human-associated bacterium showing co-migratory genetic patterns, it was shown that strict vertical transmission is surprisingly rare as well (Delport et al. 2006). Most transmission events appear to happen within families (in developed countries) or even between families (in developing countries), presumably due to different degrees of intra- and inter-family contact conditional on lifestyle (Schwarz et al. 2008). Thus, even though we observed vertical transmission of *E. rectale*, strict vertical transmission is not necessary to observe population structure shaped by physical isolation in human-associated bacteria.

We have added a Supplementary Figure as well as results and discussion text to describe the vertical transmission analysis and to put this consideration into context.

Main text: To assess the possibility of inter-individual *E. rectale* strain transmission in human populations, we further analyzed metagenomic data from mother-infant pairs in multiple cohorts (N=532 samples; **Methods**) and found evidence of vertical transmission (25% transmission rate within the first year of infant's life, **Suppl. Fig. 13**). Overall, these analyses suggest that *E. rectale* is specific to humans and that it can be transmitted within populations.

Discussion: While population structure shaped by isolation by distance has previously been described for the (opportunistic) human pathogen *H. pylori* [36–38], here we report for the first time similar evolutionary signatures in a human gut commensal. Interestingly, vertical transmission rates were found to be low in both *H. pylori* [39,40] and *E. rectale*. The estimated transmission rate of 25% observed between mother-infant pairs for *E. rectale* (**Suppl. Fig. 13**) suggests that strain seeding from the local (social) environment contributes to the observed biogeographic stratification.

Minor issues:

The statement in the abstract regarding the loss of the motility operon confirms the results of a previous study (doi: 10.15252/msb.20177589). This should be indicated in the abstract to avoid the interpretation that this is a novel finding.

We changed this to emphasize that it is not an entirely novel finding (Costea et al. mention only the absence of the largest motility operon), but references cannot be used in the abstract according to the guidelines of Genome Biology.

Abstract: We confirm that a relatively recently diverged *E. rectale* subspecies specific to Europe consistently lacks motility operons and that it is immotile *in-vitro*, probably due to ancestral genetic loss.

In the introduction, the authors characterize their study as "the first large-scale population-level genomic analysis of *E. rectale*". This is the first time MAGs have been used to study *E. rectale*, but not the first large-scale population-level genomic analysis.

We changed this to make it clear that it is the first study of such a kind that is based on genomes from metagenomes.

Introduction: The genomes that were assembled from metagenomes were used for the first large-scale genome-based population-level genomic analysis of *E. rectale* exemplifying how studies typically performed with cultured isolate sequencing data can be performed on carefully quality-controlled genomes from metagenomes.

Were the metagenomes assembled for the current study or assembled previously? This is not clear from lines 102-104.

Metagenomic assemblies of all but one dataset were already available from Pasolli et al. (2019). The remaining one was assembled for Tett et al. (2019). Starting from the binning procedure (reference-based

in this work and reference-free in Pasolli et al), the downstream computational operations are novel and performed specifically for this work. We now make this clear in the main text.

Main text: We applied this pipeline on a collection of 6,775 gut metagenomic assemblies obtained from our previous studies (Pasolli et al. 2019; Tett et al. 2019).

On lines 113-114 the authors write that their pipeline greatly increased completeness at a small cost of contamination. From Fig 1E it looks like contamination was greatly increased. Please indicate summary statistics in the main text to support the above claim or reword as necessary.

We thank the reviewer for pointing this out. We realized that the original Figure 1F was unintentionally biased against our approach. This was because we reported only bins with contamination below 5% for MetaBAT2 whereas the bins generated using our approach were not filtered for contamination. When considering only the correct set of bins, the difference in contamination between the two approaches is less pronounced, as now reported in Supplementary Figure 2.

Supplementary Figure 2: Completeness and contamination estimates for the reference-based binned used in this study and a reference-independent pipeline using metaBAT2 [2,16]. Only genomes with >90% completeness and <5% contamination in both approaches are shown.

Comparing bins between the reference-based and reference-independent (metaBAT2) binning approaches, contamination tends to increase less than completeness in reference-based binning. This is formally quantified by the combined score defined by Parks et al. (Parks et al. 2017) obtained by subtracting contamination from completeness for each bin.

This combined measure, now reported in Figure 1D and 1E, shows that our approach slightly outperforms metaBAT2.

Fig. 1D: Scatterplot of Bin scores (Defined for each bin by subtracting contamination from completeness) between reference-based bins and bins generated by metaBAT2. **Fig. 1E** Boxplot of Score differences between bins generated through reference-based binning and metaBAT2, showing that reference-based bins tend to have a higher score.

Accordingly, we have reworded the text to better reflect that the increase in score is limited. Furthermore, we have recomputed Figures 1E and 1F to only show genomes that are more than 90% complete and less than 5% contaminated in both approaches.

Main text: We found that our pipeline reconstructs *E. rectale* genomes with high fidelity, slightly outperforming reference-free metagenomic binning when considering a combined measure of estimated genome completeness and contamination [2,17] (**Fig. 1D**, **Fig. 1E**, **Suppl. Fig. 2**).

The classification of high-quality MAGs (HQ) has been misappropriated. Beyond 90% complete and 5% contamination, the community-accepted definition requires a near-complete complement of tRNAs and rRNAs (doi: 10.1038/nbt.3893).

The reviewer correctly points out that the definition for HQ MAGs by Bowers et al. also includes the presence of tRNAs and rRNAs. We acknowledge this and now point it out in the main text. We also want to mention that there appears to be a discrepancy between what Bowers et al. propose and what is done in practice, as neither of four recent, large scale metagenomic assembly efforts have considered the presence of tRNAs or rRNAs in their definition of “High Quality”/“near-complete” MAGs (Pasolli et al. 2019; Almeida et al. 2019; Nayfach et al. 2019; Almeida et al. 2019; Parks et al. 2017). This reflects the intrinsic difficulty of assembling rRNA genes, and current state of the art studies are thus not requiring this in the quality control step. We discuss this in the text as reported below .

Main text: The 1,321 HQ *E. rectale* genomes contain less than 400 contigs and passed recently proposed completeness and contamination cutoffs (90% and 5% respectively) for high-quality metagenome-assembled genomes [20]. In line with recent large-scale metagenomic assembly efforts [2,12,13], we did not consider the presence of tRNA- and rRNA genes as criteria for high quality metagenome-assembled genomes because of the inherent difficulty of reconstructing genes that are conserved across related species [20] (**Fig. 1B**).

Please indicate what was the input to the PAM clustering algorithm. Was this % DNA identity across core genes between genomes? This is an important piece of information to report.

Thanks for pointing out that this is not clear from the main text. PAM clustering was indeed performed using core gene hamming distances between genomes and we have now added it to the main text.

Main text: Clustering of core gene genetic distances using Partitioning Around Medoids (PAM) [21] supported the existence of four subspecies (Prediction Strength consistently over 0.8 for $k = 4$, **Suppl. Fig. 4, Fig. 2D, Methods**), one of which was not observed before [9–11].

What was the rationale for using the PAM algorithm? From looking at their results in 2A, there are some subspecies assignments that disagree with the topology of the tree. Why did the authors not use the tree itself to define the lineages? Additionally, why the choice of 4 clusters (as opposed to 3 or 5)? I wonder how the conclusions in the paper would change with a different # of subspecies.

It is true that in many analyses subspecies are defined directly on the phylogenetic tree. However, we chose to adopt an approach which is fully automatic, independent and agnostic with respect to the downstream analysis, and not dependent on subjective choices in order to avoid any overfitting of the data. To this end, we used the prediction strength metric on raw genetic distances together with the PAM algorithm (explained above). This approach produces the largest number of genetically well-defined groups of samples that can be consistently recapitulated over random subsamples of the data, which we use a definition of subspecies in our work.

There is in fact a high consistency between the phylogenetic tree and the subspecies defined as above. ErEurope, ErAsia, and ErAfrica are well-defined in the tree whereas ErEurasia instead has some inconsistencies. (i) 4 strains are placed between ErAfrica and ErAsia, but these strains are very highly rooted in the tree, so their subspecies assignment would be problematic also using the tree. (ii) some other strains (52 strains in total) are instead included in two subtrees that are branching off externally to both ErEurope and the great majority of ErEurasia strains. These mainly represent Fijian and Ethiopian strains which are also genetically distinct in the ordination. Based on the tree, these two groups of strains could not be attributed to ErEurasia and should define two distinct additional clades; however, separating subspecies based on this level of genetic diversity would over-separate the clade representing the rest of ErEurasia and would be inconsistent with the genetic diversity of the other three subspecies. Overall, the PAM algorithm thus produces consistent subspecies compared to the tree-based approach.

Additionally and importantly, the choice of the number of subspecies, as reported above, does not influence our conclusion regarding isolation by distance as this correlation also holds when correlating geographic and genetic distances between all pairs of strains in Eurasia and Africa.

We have added a main text mention that there are inconsistencies between the subspecies assignment and the tree topology.

Main text: Three of these four subspecies are large and well-defined monophyletic subtrees in the phylogeny and only a minority of strains of the largest subspecies is falling in divergent paraphyletic subtrees (**Fig. 2A**).

Reviewer #2

Overall this is an excellent analysis method applied to an important human commensal. The work uncovers some interesting novel biology and represents a significant advance that is of broad interest to the field.

We would like to thank the reviewer for this positive assessment.

The authors present a targeted approach to reconstruct genomes from metagenomes that is superior to "traditional" assembly based methods. Analysis of *Eubacterium rectale* genomes reconstructed from metagenomes suggests an interesting human specific, geographically distinct population structure with compelling differences in motility and carbohydrate utilisation. Technically the computational analysis is generally excellent with minor points of clarification required. The inclusion of experimental validations was positive but the manuscript would benefit substantially from greater coverage in both the figures and text of this data. Sufficient data is provided for replication of both informatic and experimental validation.

The following points should also be addressed:

1. The authors have included 6775 human gut metagenomes (including newly sequenced samples to expand the diversity of donors; however, there are now at least 10,000 such human gut metagenomic samples available in ENA/NCBI on what basis were these samples selected and what impact may that have on the results described. This is particularly relevant for assessing the robustness of the newly proposed 3-step process and any biases that may exist with sample preparation, data quality, etc.

A very recent, large scale assembly study has used almost 8000 gut metagenome samples (Pasolli et al. 2019) and was slightly extended by an even more recent integrative approach (Almeida et al. 2019). The first work indeed reports a total of almost 10,000 samples, but almost 2,000 were from body sites other than the gut in which *E. rectale* is not found. In our present study we have used the majority of those (more than 6500) plus some more samples from non-industrialized populations from another study (Tett et al. 2019). The datasets that were not included were all from already well represented populations (Europe, North America, China). We thought those were unlikely to add relevant phylogenetic/functional diversity and that they would further skew the underrepresentation of e.g. African populations. Furthermore, their inclusion would have also added substantial computational overhead (the phylogenetic analyses performed here are more computationally demanding than those in Pasolli et al. 2019).

We took here the opportunity to confirm our intuition explained above. To this end, we have extracted genomes from these originally excluded datasets for this resubmission and have clustered them genetically with the set of genomes used in our study. We found that the genomes are all within 2% genetic distance to centroids of ErEurope, ErEurasia and ErAsia (**See Figure below**), consistent with their respective intra-subspecies genetic diversity (See **Figure 2C** in the manuscript). This suggests that the genomes extracted from these additional studies would have been assigned to these three subspecies and thus very likely would not have added more meaningful information. We thus decided not to consider these datasets for the rest of the analysis.

It is also of note that in our work we considered assemblies of several non-human primate gut metagenomes published recently (Manara et al. 2019) that were not included in the previous large-scale efforts and which are not counted in the 6775 metagenomes we mention in the text.

Histogram of genetic distances between 184 HQ genomes and their closest subspecies centroid isolated from 1237 shotgun metagenomes from European, American or Chinese shotgun metagenomic cohorts not considered in this study.

2. The phylogenetic tree (Figure 2a) appears to show members of the ErEurasia subspecies co-occurring with ErAsia and ancestral to ErEurope further comment should be made to explain this relationship given the apparent inconsistency between the genetic relatedness and the phylogenetic tree.

We thank the reviewer for this comment and point out that reviewer 1 has made a very similar remark which we address above. We're copying the corresponding paragraph below:

There is in fact a high consistency between the phylogenetic tree and the subspecies defined as above. ErEurope, ErAsia, and ErAfrica are well-defined in the tree whereas ErEurasia instead has some inconsistencies. (i) 4 strains are placed between ErAfrica and ErAsia, but these strains are very highly

rooted in the tree and with low support, so their subspecies assignment would be problematic also using the tree. (ii) some other strains (52 strains in total) are instead included in two subtrees that are branching off externally to both ErEurope and the great majority of ErEurasia strains. These mainly represent Fijian and Ethiopian strains which are also genetically distinct in the ordination. Based on the tree, these two groups of strains could not be attributed to ErEurasia and should define two distinct additional clades; however, separating subspecies based on this level of genetic diversity would over-separate the clade representing the rest of ErEurasia and would be inconsistent with the genetic diversity of the other three subspecies. Overall, the PAM algorithm thus produces consistent subspecies compared to the tree-based approach.

We have added a main text mention to explain that there are inconsistencies between the phylogenetic tree and genetic clustering.

Main text: Three of these four subspecies are large and well-defined monophyletic subtrees in the phylogeny and only a minority of strains of the largest subspecies is falling in divergent paraphyletic subtrees (**Fig. 2A**).

3. The authors state genomes assembled from non-human primates were not closely related to *E. rectale*, were these samples subjected to the same analysis performed on the human samples? This would provide the most compelling evidence and should be included and/or stated specifically.

These metagenomes were assembled in the same way, but indeed originally binned using metaBAT2.

However, we agree with the reviewer that the same exact procedure should have been used. So, in order to confirm that this result does not represent an artifact caused by the binning method, we have generated *E. rectale* genomes from these non-human primate metagenomic assemblies using the reference-based binning approach used in this study. None of the extracted genomes are more than 5% complete, corroborating our previous result. We now describe this result in the methods section.

Methods: In order to find *E. rectale* genomes assembled from wild non-human primate metagenomes, we assembled and binned as described elsewhere [2,51] a total of 2,895 metagenomic high-quality genomes obtained from 175 publicly available metagenomes from wild, non-human primates (**Suppl. Table 7**). These 175 metagenomes come from four different datasets spanning 22 non-human primate species including Chimpanzees and Gorillas from 14 different countries on five continents [45–48]. We then estimated genetic distances between each of the reconstructed genomes and the set of *E. rectale* isolate genomes using MASH [52], and found that not a single bin generated from the non-human primates was within 23% genetic distance of any *E. rectale* isolate. To confirm that this result is not dependent on the binning method, we also applied the reference-based binning procedure we proposed in this work to these assemblies. We found that not a single bin was more than 5% complete, confirming our previous result that the metagenomic assemblies of wild non-human primates used in this study do not contain *E. rectale* genomes.

4. Have the authors undertaken an assessment of the source studies from which the isolates originate? How many different studies contribute to each subspecies and could study related bias impact the observed results and/or apparent prevalence?

Isolate genomes are coming from six different sources in Europe, North America and China. Isolate genomes were isolated for three of the four subspecies; none of the isolate genomes used in this study belong to ErAfrica. Given the very small genetic variability of subspecies (no pair of genomes has a genetic distance of >3%) and the fact that we have many more manually curated genomes from a geographically diverse set of studies that were also used as reference genomes in the extraction, it is reasonable to assume that the choice of isolate genome set does not influence observed results or apparent prevalences of subspecies.

5. Given geographical separation is proposed as the driver of speciation the (relatively) limited number of functional differences identified is surprising. The specific order of analysis was unclear, were both presence/absence and sequence based selection considered for all genes? It may be of value to place additional focus on sequence based analysis in addition to differential abundance to determine if the inconsistencies observed within subspecies can be explained by sequence variation, etc.

We are not entirely sure if we understand what the reviewer meant by 'sequence based selection'/'sequence variation' in the context of functional annotation. The functional differences were inferred using representative sequences of gene clusters generated using roary. The gene sequences at 95% sequence identity, making sure not to discard too much sequence variation. In our opinion, given the relatively small absolute degree of genetic variation, the limited functional differences are not surprising.

6. Could the authors speculate on the relevance/selective forces driving the loss of motility associated genes within this subspecies and the associations with carbohydrate metabolism? The rationale behind looking for carbohydrate metabolism

ErEurope strains might have been forced to change and extend their repertoire of catabolic carbohydrate active enzymes to be able to metabolize a wider range of energetically unfavourable carbohydrates such as Inulin and Xylan due to the inability to scavenge for energetically more favourable carbohydrates.

We have added this text into the discussion in order to make our thought process clearer.

Discussion: We speculate that, with the lack of motility, ErEurope strains might have been forced to change and extend their repertoire of catabolic carbohydrate active enzymes to be able to metabolize a wider range of energetically unfavourable carbohydrates such as Inulin and Xylan (**Tab. 1**) due to the inability to scavenge for energetically more favourable carbohydrates.

7. Experimental validation is an important and valuable addition to papers of this nature. I suggest further emphasis and analysis of the results should be included in the main text and figures for both the motility and carbohydrate utilisation.

We have moved the Supplementary Table (the in-vitro motility assays for six isolates) to Figure 4. We moved Figure 4D and Figure 4E into the supplement. We also added the results of the in-vitro carbohydrate utilization assay as a main table and have put it into the main text.

Minor

1. The use of metaSPAdes with paired end and MegaHIT with unpaired reads could be mentioned in the main text for clarity.

We have changed the main text to make this clear.

Main text: We applied this pipeline on a collection of 6,775 gut metagenomic assemblies obtained from our previous studies [2,14]. These assemblies were generated using metaSPAdes [15] if paired-end reads were available or MegaHIT otherwise [16].

2. Inclusion of terminology such as polymorphism based heterogeneity assessment on Line 129 rather than just referring readers to the methods would be preferable.

We have added a brief statement to the main text that the method is based on polymorphic site rate estimation across core genes.

Main text: The genomes were however further required to pass an additional quality measure we developed based on polymorphic site rates across core genes to flag genomes that are likely to incorporate strain-level variation from more than one strain (**Methods**).

3. Separation of Figure 1 into two figures should be considered to improve clarity, at a minimum separation for Figure 1A

We have moved Figure 1A to the supplement.

4. The cultured isolates should be highlighted on the phylogenetic tree in Figure 2A

We have added markers for isolate genomes to the phylogenetic tree in Figure 2A.

5. Colours on Figure 2A could be improved to more clearly separate colours for subspecies, continent and country. Also consistency of colours between sub-part of figures (e.g. 2A and 2B) would make the patterns clearer to the reader

We have adopted the color coding for countries from Figure 2B for Figure 2A, producing consistent and clearer color coding.

6. The authors should carefully choose their wording regarding the association with an ancient link as the presented data and phylogenetic dating does not seem to prove this conclusively.

We thank the reviewer for this remark. In the context of the isolation by distance hypothesis, we have reworded large parts of the results- and discussion sections. We think that we now adequately articulate caveats and limitations of this aspect of the study.

Second round of review

Reviewer 1

I thank the authors for a thorough response to my questions. My only remaining issues are in regard to bin quality.

I see that the authors have removed panel 1A, which indicated that the MCHQ bins were 98% complete. I'm still a little confused how these can be 98% complete but more than 1 Mb smaller than isolates. Maybe the completeness was estimated before manual curation?

Regarding the comparison of the reference-guided vs reference-independent approaches - I don't think there's enough evidence to claim that the reference-independent approach is better. It clearly results in higher estimated completeness, but at a cost of higher estimated contamination. Maybe this approach is useful when prioritizing sensitivity over specificity?

The authors made several changes to the text and figures which I think have hidden this result. I'd prefer if the authors reported the results as they did in the first version: "We found that this pipeline reconstructs *E. rectale* genomes with high fidelity, outperforming reference-free metagenomic binning in terms of completeness [2,16] while slightly increasing contamination". In addition, please report the mean/median % change in completeness and the % change in contamination.

The current version reads "We found that our pipeline reconstructs *E. rectale* genomes with high fidelity, slightly outperforming reference-free metagenomic binning when considering a combined measure of estimated genome completeness and contamination". This "combined measure" was obtained by subtracting contamination from completeness for each bin. While the authors say that this was the same approach as used in Parks et al 2017, this is not quite accurate. The Parks et al. quality score is defined as: completeness - 5 x contamination (assigning contamination a large negative weight). I wonder if the authors claim will stand after this adjustment.

Reviewer 2

The authors have addressed the questions and concerns raised previously.

The revised manuscript is substantially improved. All the necessary clarifications and caveats have been incorporated to enable appropriate interpretation and provide context for the manuscript.

Authors Response

Point-by-point responses to the reviewers' comments:

Reviewer #1: I thank the authors for a thorough response to my questions. My only remaining issues are in regard to bin quality.

I see that the authors have removed panel 1A, which indicated that the MCHQ bins were 98% complete. I'm still a little confused how these can be 98% complete but more than 1 Mb smaller than isolates. Maybe the completeness was estimated before manual curation?

Sorry for the misunderstanding, but we just moved Figure 1A to the supplement rather than having removed it. We confirm that completeness was estimated *after* manual curation. We hypothesize that in the samples in which *E. rectale* was very highly abundant and that were used for the MCHQ bins, there were some binning biases leaving out some contigs. However, these contigs did not affect completeness nor contamination. Moreover and most importantly, considering the purpose of the MCHQ bins (which we reiterate were used for the reference-based binning and not for the genome analysis), this is not something that would affect contamination of the retrieved bins, and did not affect completeness either.

Regarding the comparison of the reference-guided vs reference-independent approaches - I

don't think there's enough evidence to claim that the reference-independent approach is better. It clearly results in higher estimated completeness, but at a cost of higher estimated contamination. Maybe this approach is useful when prioritizing sensitivity over specificity?

As the reviewer correctly points out, the difference in performance between the two approaches is limited and - to an extent - not completely objectively quantifiable as it comes down to a tradeoff between sensitivity and specificity. Sensitivity is clearly key in our study, and so we think we have at least evidence that the reference-guided approach is not worse than the reference-independent approach.

We edited the claim as reported here and we agree this is a more fair statement:

“We found that this pipeline reconstructs *E. rectale* genomes with high fidelity, outperforming reference-free metagenomic binning in terms of completeness [2,17] while slightly increasing contamination (1,7% median increase in completeness, 0.5% median increase in contamination) (**Fig. 1 D, Fig. 1 E**). The pan-genome characteristics of the reconstructed *E. rectale* genomes more closely resemble those of isolate *E. rectale* genomes than the *E. rectale* genomes coming from reference-free binning (**Fig. 1 F, G**), further suggesting that they generally are of high quality.”

The authors made several changes to the text and figures which I think have hidden this result. I'd prefer if the authors reported the results as they did in the first version: "We found that this pipeline reconstructs *E. rectale* genomes with high fidelity, outperforming reference-free metagenomic binning in terms of completeness [2,16] while slightly increasing Contamination".

In addition, please report the mean/median % change in completeness and the % change in contamination.

As per the reviewers request, we reverted the phrasing as well as Figure 1D, E to the initial versions and now also report median % change in completeness and contamination between our reference-based binning approach and the approach used in (Pasolli et al. 2019) . Furthermore, we have removed Supplementary Figure 2 to be consistent with the previous version of Figure 1 as requested.

The current version reads "We found that our pipeline reconstructs *E. rectale* genomes with high fidelity, slightly outperforming reference-free metagenomic binning when considering a combined measure of estimated genome completeness and contamination". This "combined measure" was obtained by subtracting contamination from completeness for each bin. While the authors say that this was the same approach as used in Parks et al 2017, this is not quite accurate. The Parks et al. quality score is defined as: completeness - 5 x contamination (assigning contamination a large negative weight). I wonder if the authors claim will stand after this adjustment.

We apologize for this mistake, the reviewer is correct in that we used a different formulation of the score metric proposed in Parks et al. The weight of contamination with respect to completeness is however quite arbitrary in this and other scores, we think that by reverting back the corresponding section in the main text to a descriptive report of the data (“ *We found that this pipeline reconstructs *E. rectale* genomes with high fidelity, outperforming reference-free metagenomic binning in terms of completeness [2,17] while slightly increasing contamination (1,7% median increase in completeness, 0.5% median increase in contamination)*“), we are comparing the two binning methods in a fair manner and are not mis-representing the data.

Reviewer #2: The authors have addressed the questions and concerns raised previously. The revised manuscript is substantially improved. All the necessary clarifications and caveats have been incorporated to enable appropriate interpretation and provide context for the

manuscript.

We thank both reviewers for their constructive comments.